# Droplet-based mechanical transducers modulated by the symmetry of wettability patterns

Luanluan Xue[1,2], An Li ⊕[1], Huizeng Li ⊕[1] ✉, Xinye Yu[1,2], Kaixuan Li[1], Renxuan Yuan[1,2], Xiao Deng[1,2], Rujun Li[1,2], Quan Liu[1,2] & Yanlin Song ⊕[1,2,3] ✉

Asymmetric mechanical transducers have important applications in energy harvesting, signal transmission, and micro-mechanics. To achieve asymmetric transformation of mechanical motion or energy, active robotic metamaterials, as well as materials with asymmetric microstructures or internal orientation, are usually employed. However, these strategies usually require continuous energy supplement and laborious fabrication, and limited transformation modes are achieved. Herein, utilizing wettability patterned surfaces for precise control of the droplet contact line and inner flow, we demonstrate a droplet-based mechanical transducer system, and achieve multimodal responses to specific vibrations. By virtue of the synergistic effect of surface tension and solid-liquid adhesion on the liquid dynamics, the droplet on the patterned substrate can exhibit symmetric/asymmetric vibration transformation when the substrate vibrates horizontally. Based on this, we construct arrayed patterns with distinct arrangements on the substrate, and employ the swarm effect of the arrayed droplets to achieve three-dimensional and multimodal actuation of the target plate under a fixed input vibration. Further, we demonstrate the utilization of the mechanical transducers for vibration management, object transport, and laser modulation. These findings provide a simple yet efficient strategy to realize a multimodal mechanical transducer, which shows significant potential for aseismic design, optical molding, as well as micro-electromechanical systems (MEMS).

From pistons driving gears in an engine to muscles pulling bones in an animal's body, mechanical energy transfer and conversion systems are ubiquitous in our daily lives, and act as one of the most fundamental and widely studied fields. Especially, transforming a symmetric input into an asymmetric output endows the mechanical transducer with the potential to guide, damp, and control signals and energies[1,2], which is widely used in vibration manipulation[3], wave modulation[4], micro-mechanics[5], and mechatronics[6]. To realize asymmetric output, active and passive mechanical transduction strategies are usually employed.

The active strategy uses robotic mechanical metamaterials to break the symmetric interaction between the building blocks, usually requiring continuous energy supplement[7]. For the passive strategies, the deformation symmetry of materials is broken by asymmetric microstructures[8–11] or internal orientation[1], while laborious fabrication is usually involved and output modes are limited.

Droplets, though typically considered as liquid instead of material, are actually a kind of metamaterial[12,13] and showcase the advantages of easy accessibility, component diversity, and volume

---

[1]Key Laboratory of Green Printing, CAS Research/Education Center for Excellence in Molecular Sciences, Beijing National Laboratory for Molecular Science, Institute of Chemistry, Chinese Academy of Sciences, Beijing 100190, China. [2]University of Chinese Academy of Sciences, Beijing 100049, China. [3]Xiangfu Laboratory, Jiashan, Zhejiang 314102, China. ✉e-mail: lihz@iccas.ac.cn; ylsong@iccas.ac.cn

tunability, which might possess the potential for mechanical transformation. Through rational design, droplets can exhibit various appealing metamaterial-possessed properties[14], showing unique applications in optical modulator[15,16], non-Hookean spring[17], and hydrodynamical rectifier[18]. In addition, the surface tension and the flowability of the liquid inside the droplet endow it with bizarre mechanical properties different from traditional solid materials, which makes it widely used in micromechanics[19–21]. Especially, the heterogeneous treatment of the substrate surface has greatly improved the diversity, complexity, and control-precision of the droplet behaviors on it[22–24], as evidenced by recent advances including droplet gyrational bouncing[25], self-splitting[26], and polygonal spreading[27,28], etc., making it an excellent candidate for unique functionalities.

Herein, we exhibit a droplet-based mechanical transducer utilizing wettability patterned surfaces for precise control of the droplet vibration behavior, and achieve multimodal output from specific vibration input. We demonstrate that when the substrate vibrates horizontally, the deformation symmetry of the droplet can be regulated by the pattern symmetry, and the symmetric input can be transformed into asymmetric output. Based on this, we construct arrayed patterns with distinct arrangements on the substrate, and employ the swarm effect of arrayed droplets to achieve three-dimensional and multimodal actuation of the target plate. The capability of the droplet-based mechanical transducers to enhance, damp, and guide vibration is verified by applications in aseismic vibration management, object transport, and laser modulation.

## Results and discussion

Vibrations are usually transformed into other forms of motions and energy for output through the use of mechanical transducers[29,30]. Here we place/adhere different materials on a horizontally-vibrating stage to compare their vibration transformation abilities (Fig. 1a, see experimental setup in Supplementary Fig. 1). The vibration stage is superhydrophobic with superhydrophilic patterns, showing physical homogeneity and chemical heterogeneity (Supplementary Fig. 2 and Supplementary Fig. 3). The vibration curve of the stage is shown in Fig. 1a. Although the input vibration mode is fixed, the actuation behaviors of different materials are highly diverse. As schemed in Fig. 1b (left), when a steel bead, as a rigid-body, is adhered to the stage, it shows no deformation with its height unchanged, exhibiting fully synchronous dynamics with the vibrating stage (Fig. 1c, the blue curve), indicating that the steel bead cannot transform the vibration. By comparison, when a hydrogel bead, the typical soft-matter, is adhered to the stage, it swings around the adhesion point and shows symmetric deformation (Fig. 1b, right). When the vibrating stage deviates away from the equilibrium position, the hydrogel bead is stretched in length and compressed in height. As the vibration signal is sinusoidal, the bead shows deformation symmetry, and transforms the input vibration into symmetric height fluctuation (Fig. 1c, the green curve). However, when we place a 10 µL glycol droplet on the vibrating stage, the droplet is elongated (Fig. 1d) and shaped by the pattern, which contains two circles of unequal sizes and a connecting channel, as demonstrated in detail in Supplementary Fig. 4. The distribution of the liquid between the two circles reaches hydraulic equilibrium under the Laplace pressure, with most of the liquid stored in the large circle. The height of the static droplet is recorded as $h_0$. When vibrating, the liquid fluctuates between the circles under synergistic effects of the capillary force and the inertia force. The droplet height $h$, defined as the height difference between the droplet vertex and the substrate during the whole vibration period, is recorded. Specifically, when the stage vibrates towards the left, part of the liquid flows into the right small circle due to the inertia force. As a result, the droplet is compressed to a height smaller than $h_0$. Similarly, when the stage vibrates towards the right, the liquid flows into the left big circle, showing a height larger than $h_0$. The variation of the droplet height is

summarized in Fig. 1e and Supplementary Video 1. It clearly shows that when the stage vibrates in opposite directions, the response heights of the droplet are unequal at the two ends, indicating that the droplet shows deformation asymmetry, with the ability to transform the symmetric input into asymmetric output.

To investigate the influence of pattern design on the transformation symmetry of the droplet-based vibrating system, three types of patterns, including circle, dumbbell, and gourd, are fabricated on the stage. Here we characterize the transformation symmetry of the system using the symmetry of droplet height variation, with the asymmetric coefficient ($\eta$) equal to the nondimensionalized height difference of the droplet when the stage locates at the leftmost and rightmost positions: $\eta = \left| h_{\text{leftmost}} - h_{\text{rightmost}} \right| / h_0$. When the droplet is placed on the circle-patterned stage and the vibration is applied, the droplet shows a typical bending mode behavior, with the contact line pinned by the pattern and its cap swinging around symmetrically (Supplementary Fig. 5, and Supplementary Video 2). The dependence of the droplet height on the stage displacement is plotted in Fig. 2a. The asymmetric coefficient $\eta$ equals to 0 for the droplet vibrating on the circle pattern, meaning that the transformed vibration is still symmetric. Similar results are obtained using a dumbbell pattern composed of two equal-sized circles and a rectangle channel (Fig. 2b), with $\eta$ being 0. The morphology of the droplet is shown in Supplementary Fig. 6 and Supplementary Video 2. Although the liquid in the droplet fluctuates between the two circles, the dumbbell pattern still induces symmetric liquid flow and symmetric output.

Therefore, to transform the symmetric input into asymmetric output, the symmetry of the pattern, and essentially the symmetry of the liquid flow, need to be broken. Thus, we design a gourd pattern (Supplementary Fig. 4) using two different-sized circles with a trapezoid connecting channel (see the design principle in Supplementary Information 2.1). Due to the Laplace pressure, the liquid mainly lies on the large circle in a static state. When the stage vibrates towards the left, the liquid shunts into the small circle at the right, showing a squeezed morphology and a reduced height. In the subsequent half cycle, the stage vibrates towards the right, and the liquid accumulates into the large circle at the left, exhibiting an expanded height. Obviously, the system exhibits different vertex heights at the leftmost and rightmost positions (Fig. 2c), showing an asymmetric coefficient $\eta$ of 0.23, indicating its asymmetric output.

Subsequently, the influence of the gourd pattern design on the transformation asymmetry of the system is investigated. For gourd patterns with different diameter ratios, we find that the asymmetric coefficient reaches the maximum at a diameter ratio of 0.67 (Fig. 2d). This can be explained that for a gourd pattern with a small diameter ratio (such as 0.17), the small circle cannot effectively shunt the liquid, causing limited height variation. When the diameter ratio approaches 1, the asymmetry of the pattern is weakened, which also leads to a reduced height variation. Besides the pattern design, the asymmetric coefficient $\eta$ can also be modulated by vibration parameters, including frequency and amplitude. The dependence of $\eta$ on the vibration frequency shows a complex trend, which reaches the peak at 30 Hz (Fig. 2e). This is probably because the droplets resonate at specific frequencies, and the wettability pattern alters the intrinsic frequency of the droplet. By contrast, the asymmetric coefficient exerts a simple and monotonically increasing relationship with the vibration amplitude, until reaching the maximum of 0.24 (Fig. 2f). Further enlarging the amplitude will cause the overviolent flow of liquid, and the effect of the pattern asymmetry is weakened and its asymmetric coefficient is decreased. In our following tests, the diameter ratio of the gourd pattern is fixed at 0.5, and the frequency and amplitude of the stage are 30 Hz and 350 µm unless specifically stated.

Then we explore the swarm behavior of the vibrating droplets, which may be more engaging and diverse. As the translation along and rotation around $x, y, z$-axis are the basic motions in a three-dimensional

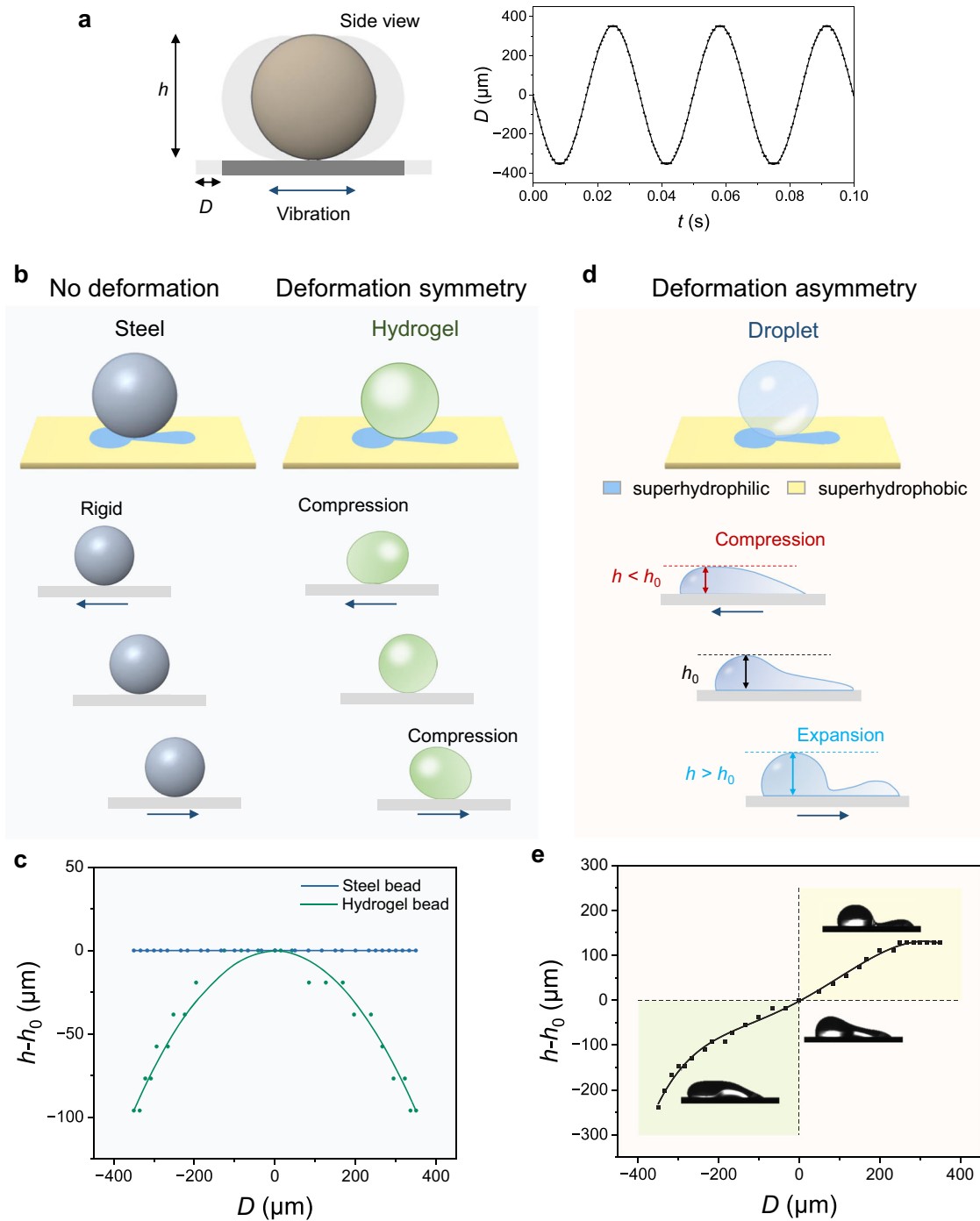

**Fig. 1 | Vibration transformations of a steel bead (rigid-body), a hydrogel bead (soft-matter), and a droplet on a patterned stage. a** The scheme of the vibration test. The vibration stage is driven by a sinusoidal signal, with a frequency of 30 Hz and an amplitude of 350 μm. Here $D$ is the displacement of the stage. **b** The no deformation response of the steel bead and the symmetric deformation response of the hydrogel bead to the stage vibration. **c** The height variation of the steel and hydrogel beads at different displacements of a vibrating cycle. **d** The asymmetric deformation response of the droplet. **e** The height variation of the droplet during the vibration. The insets are the pictures of droplet morphology during vibration.

world and can be integrated to achieve more complex motions, we attempt to achieve these six modes using the swarm behavior of droplet-based mechanical transducers. A three-layered stage-droplets-plate structure is constructed to transfer the swarm behavior of multi-droplets to the upper plate, as shown in Fig. 3a and Supplementary Fig. 7. When the stage is driven by a sinusoidal wave along $x$-axis, the behavior of the upper plate is synergistically driven by four droplets, and is mainly influenced by the pattern on the stage. We classify the swarm behavior of the droplets into translation and rotation, and

accordingly design arrayed patterns of different arrangements, including translational-symmetric arrangement and translational-asymmetric arrangement. To realize translation, the four droplets must exert a directional force on the upper plate that passes through its center of mass. Therefore, we design a translational-symmetric arrayed pattern on the stage to make the droplets move in-phase. Meanwhile, the droplet behavior needs to be symmetric to make the upper plate translate along the $x$- and $y$-axis, and have asymmetric height variation to achieve the translation along the $z$-axis. On the

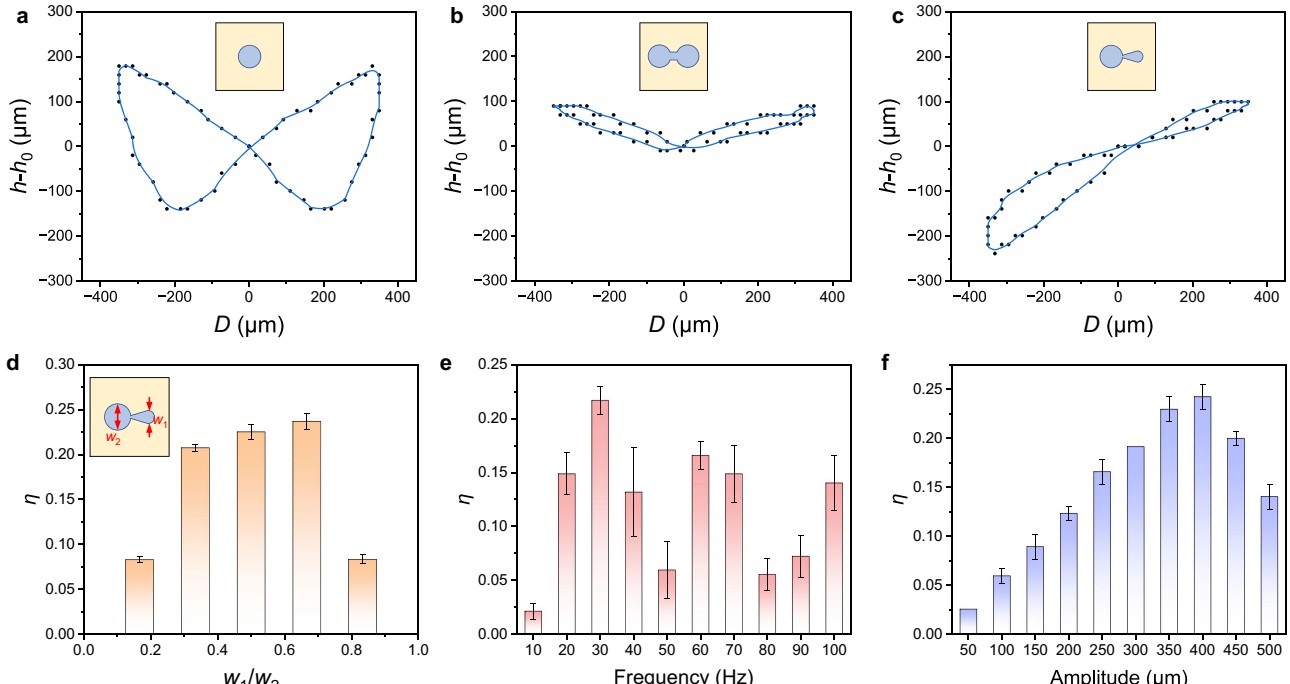

**Fig. 2 | The influencing factors of the transformation asymmetry. a** The variation of the droplet height on the circle pattern is symmetric. When the stage deviates to the largest displacement, the height reaches maximum, with $h_{leftmost} = h_{rightmost}$. **b** The variation of the droplet height on the dumbbell pattern is symmetric. The liquid flows along the pattern and the droplet vertex fluctuates between the two circles. The peak of the curve arises at the largest displacement of the stage, with $h_{leftmost} = h_{rightmost}$. **c** The height variation of the droplet on the gourd pattern is asymmetric. The lowest and highest points occur at the leftmost and rightmost positions, respectively. **d** The influence of the diameter ratio between the large and small circles on the asymmetric coefficient. $W_1$ and $W_2$ are the diameters of the small and large circles, respectively. **e–f** The influence of (**e**) vibration frequency and (**f**) amplitude on the asymmetric coefficient (The error bars represent the standard deviation, $n = 5$). Unless otherwise stated, all the droplets used in the tests are glycol with a volume of 10 μL, and the stage vibration is horizontal and is driven by a sinusoidal wave.

other hand, if the pattern is translational-asymmetric, the droplets on the stage exhibit out-of-phase motion and assert forces of different directions on the upper plate, so a torque is generated, which results in the rotation of the upper plate. Among them, the droplet motion with asymmetric height variation makes the upper plate tilt around the $x$- or $y$-axis, and the symmetric droplet motion enables the upper plate to rotate around the $z$-axis while remaining horizontal. The design principle of the arrayed patterns is summarized in Fig. 3b.

Guided by these principles, we show how a specific vibration is transformed into six modes of movement using different patterns. The translation of the upper plate along the $x$-axis is easy to achieve by pinning four droplets with superhydrophilic circles (Fig. 3c). When the stage vibrates, the droplets exhibit a synchronized bending mode, causing the plate to vibrate along the $x$-axis. Interestingly, the amplitude of the plate is larger than the stage, and can be adjusted by the pattern design. The upper plate also has slight movement along $y$ and $z$-axis, but considering their small amplitudes compared with that along $x$-axis, we focus on the $x$-axis translation as its main motion (Supplementary Fig. 8). Utilizing a heavier upper plate can further decrease its movement along $z$-axis (Supplementary Fig. 9). To make the upper plate translate along the $y$-axis, it is necessary to induce the liquid flow perpendicular to the vibrating direction. Therefore, a translational-symmetric ellipse array with 45° deflection is employed, as shown in Fig. 3d. When the stage vibrates towards the left, the liquid inside each droplet synchronously flows towards top-right, and vice versa. This induces a $y$-axis translation of the upper plate, although compounded with the vibration along $x$-axis as the oblique motion. In addition, its vibration along $z$-axis is relatively small and can be ignored. The $z$-axis translational swarm motion requires asymmetric vibrating droplets with translational-symmetric arrangement.

Therefore, we fabricate a gourd-arrayed pattern on the stage, as schemed in Fig. 3e. When the stage vibrates, the droplets height varies synchronously, making the upper plate translate along the $z$-axis. Due to the $x$-axis translation of the stage, the plate unavoidably has translation along $x$-axis, but its amplitude is significantly diminished. The translation amplitude along $y$-axis is very small and can be ignored, so we mainly focus on its translation along $z$-axis. The quantitative analysis of vibration amplitudes along $x$, $y$, $z$-axis for the above modes is shown in Supplementary Fig. 8.

To achieve the rotation of the upper plate around the $x$-axis, we arrange four gourd patterns in a rotational-symmetric way (R-gourd array for short), as shown in Fig. 3f. When the stage vibrates towards the right, the front two droplets are lower than the rear two droplets, inclining the upper plate towards the front, and vice versa. Therefore, the upper plate tilts around the $x$-axis. Similarly, four gourd patterns arranged in a mirror-symmetric array (M-gourd array) can make the upper plate tilt around the $y$-axis (Fig. 3g). To generate a rotation around the $z$-axis, the upper plate needs to remain horizontal and be subjected to a torque, so we use a mirror-symmetric ellipse array (M-ellipse array) to induce the vibration of droplets. When the stage vibrates to the left, the liquid in the left two droplets flows towards top-right, while the liquid in the right two droplets flows towards bottom-right, making the upper plate rotate clockwise. When the stage vibrates to the right, the upper plate rotates anti-clockwise, as shown in Fig. 3h. Details of these swarm behaviors are shown in Supplementary Video 3. Besides the alternating rotation shown in Fig. 3h, unidirectional and continuous rotation can be realized by designing patterns on both the stage and the upper plate, as shown in Supplementary Fig. 10 and Supplementary Video 4. Detailed discussion is provided in Supplementary Information 2.2.

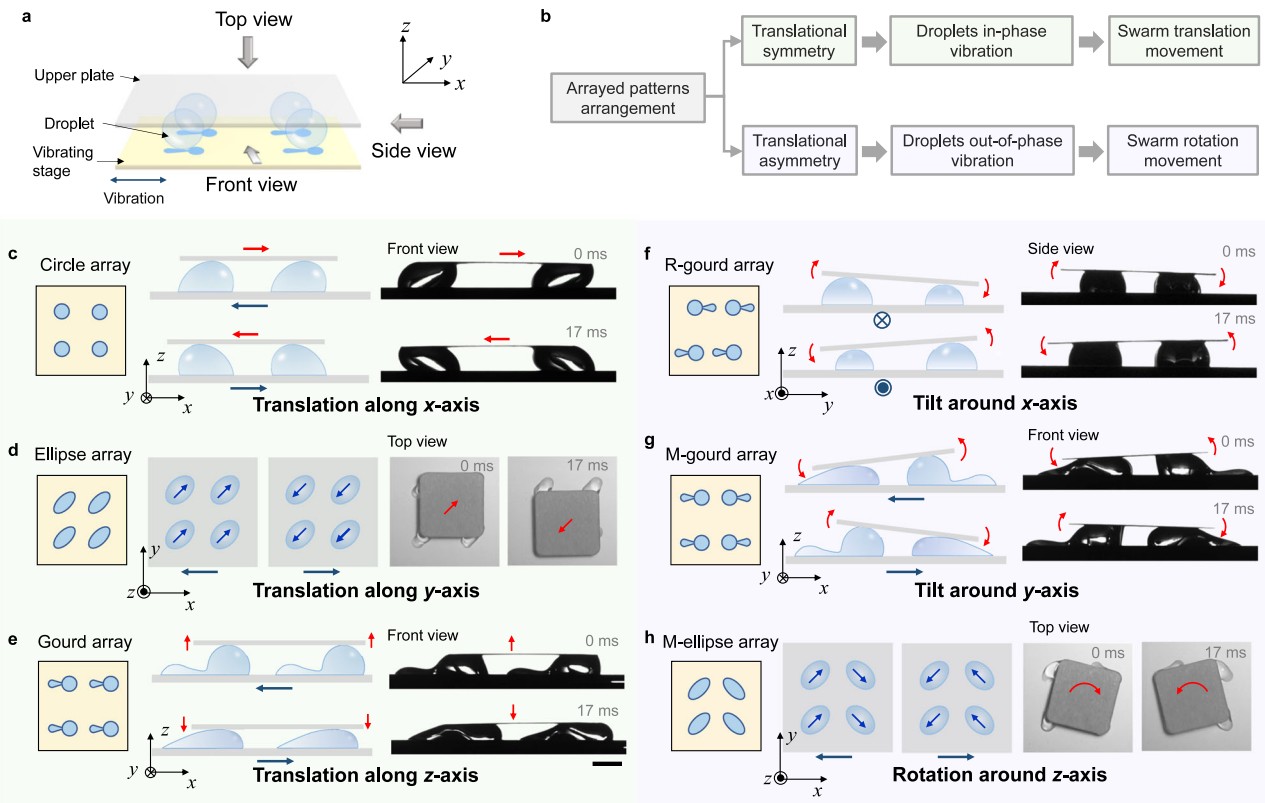

**Fig. 3 | Three-dimensional multimodal swarm behaviors of the droplet array.** **a** Scheme of the experimental setup. **b** Design principle of the wettability pattern. **c** The translational swarm behavior along *x*-axis realized using circle array. **d** The translational swarm behavior along *y*-axis realized using ellipse array. **e** The translational swarm behavior along *z*-axis realized using gourd array. **f** The tilting swarm behavior around *x*-axis realized using R-gourd array. **g** The tilting swarm behavior around *y*-axis realized using M-gourd array. **h** The rotational swarm behavior around *z*-axis realized using M-ellipse array.

Through the design and arrangement of patterns on the stage, a single horizontal vibration can be transformed into multiple modes of output. Therefore, we can use the diverse swarm behaviors of the droplets to convert, enhance, and damp the vibration energy. It is known that in an earthquake, the horizontal shock is more devastating than the vertical vibration. Here, we design circle- and gourd-arrayed patterns on the stage to demonstrate the vibration manipulating capability of the droplet-based mechanical transducers. A three-layer architecture is designed to mimic a building. As a control test, we place the architecture directly on the vibrating stage (scheme in Fig. 4a) and gradually increase the amplitude. We find that the architecture collapses at an amplitude of $276 \pm 38$ μm, due to the shear force between the layers (Fig. 4a). When we design circle-arrayed patterns on the vibrating stage and place the architecture on the upper plate, the horizontal vibration of the upper plate is enhanced compared with the stage (as shown in Supplementary Fig. 11), and the architecture collapses at an input amplitude of $160 \pm 26$ μm, showing reduced stability (Fig. 4b). By contrast, if the gourd-arrayed pattern is employed, the horizontal vibration of the upper plate shows reduction and the structure collapses at an input amplitude of $481 \pm 93$ μm, which is 1.8 times of that directly on the stage and three times of that using the circle array (Fig. 4c, Supplementary Fig. 11, Supplementary Video 5). The significantly enhanced seismic resistance performance can be attributed to the excellent horizontal-to-vertical motion conversion capability and the intense energy dissipation caused by the inner flow of the droplet-based mechanical transducer, allowing the upper plate to exhibit a remarkably-weakened horizontal vibration. The results highlight the significance of pattern design on the vibration manipulation and controllable energy output.

Moreover, the symmetric and asymmetric droplet deformation can be integrated into one stage to achieve other forms of swarm dynamics. The circle-gourd array is taken as an example, as shown in Fig. 4d. When the stage starts to vibrate, the droplets on circular patterns show slight height variation, while those on gourd patterns show obvious height variation, tilting the upper plate unidirectionally that can be used for object transportation, as shown in Fig. 4d. By comparison, the upper plate on the M-gourd array can only tilt symmetrically around *y*-axis, making the object located at the central region of the upper plate (Fig. 4e). Comparison of the transportation processes are demonstrated in Fig. 4f and Supplementary Video 6.

Furthermore, the multimodal vibration transformation of the droplet-based mechanical transducers also facilitates its application in optical modulation. Precisely steering a laser beam is fundamental for optical applications such as Lidar[31], laser micromachining[32], and optical tweezers[33]. Here we build a laser modulation system employing the droplet swarm behaviors. Figure 5a illustrates the experimental setup, which includes a laser source, a mirror reflector, a modulator, and a background panel. As shown in Fig. 5b, when the incident laser beam is cast onto the reflector, the motion of the mirror reflector drives the reflected beam to form a loop trajectory. Then, we demonstrate how to modulate the laser trajectory by altering the pattern and the vibration frequency. Theoretically, the reflected beam is static when the mirror reflector vibrates in the *xy*-plane, and only moves when the mirror reflector translates along *z*-axis or rotates around *x*-/*y*-axis. Among them, the translation along *z*-axis and the rotation around *y*-axis would cause a vertical deviation of the reflected beam, while the rotation around *x*-axis corresponds to an oblique deviation, as shown in the theoretical trajectories in Fig. 5c–h.

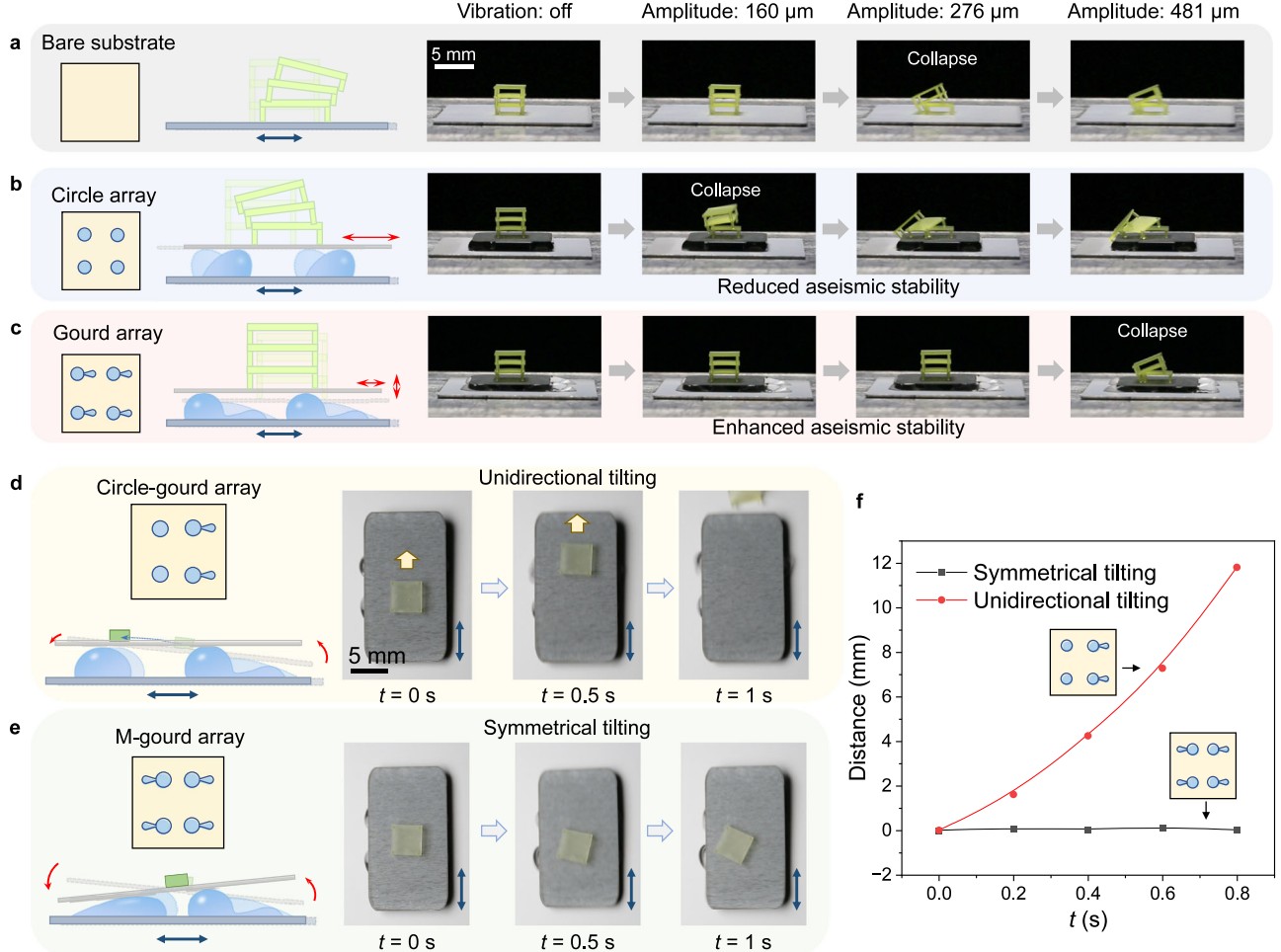

**Fig. 4 | Demonstration of the vibration manipulation and object transportation using the droplet-based mechanical transducers. a** The three-layer architecture is placed on the bare stage and collapses when the vibration amplitude is 276 μm. **b** Using a mechanical transducer with a circle array, the horizontal vibration is enhanced and makes the architecture collapse at a vibration amplitude of 160 μm. **c** The vibration is weakened by the gourd-arrayed mechanical transducer, which stabilizes the architecture and increases the critical amplitude to 481 μm. **d** Integrating the symmetric and asymmetric droplet deformation into a mechanical transducer using a circle-gourd array can transport the object directionally. **e** Symmetric tilting of the upper plate to localize the object using a M-gourd array. **f** The actuation ability comparison of the mechanical transducers.

However, as the vibration of the upper plate is not pure, the reflected beam can be actuated by all the patterns, generating complex and unique trajectories. Meanwhile, the trajectories are intensely affected by the vibration frequency of the stage. For example, when using a circle array (scheme in Fig. 5c), the reflected beam is concentrated into a laser spot at 10 Hz vibration, due to the relatively stable translation along $x$-axis of the upper plate. When the frequency increases, slight random disturbances occurred to the upper plate, causing the reflected beam to form an irregular trajectory. At a larger frequency, the upper plate fluctuates up and down periodically, which induces a linear trajectory of the reflected beam (Fig. 5c). The laser trajectories in a broader frequency range are shown in Supplementary Fig. 12, Supplementary Video 7 and discussed in Supplementary Information 2.3. For other patterns, such as the ellipse/gourd array, the M-gourd/M-ellipse array, the trajectories are vertical lines, with their lengths varying according to the frequency (Fig. 5d–g). Interestingly, the upper plate on the R-gourd pattern tilts around $x$-axis, resulting in a horizontal component of the trajectory. Thus, the trajectory emerges to be oblique (Fig. 5h). In addition, the vibration system can modulate multiple laser beams simultaneously, as shown in Fig. 5i, j. Compared with traditional laser steering systems, our approach avoids bulky systems with complex mechanical components. With simple one-

dimensional vibration input, we can obtain completely different laser trajectories by changing the pattern and frequency, demonstrating broad application prospects in multimodal optics and imaging.

In summary, we reveal that modulating pattern symmetry can regulate the vibration transformation symmetry of the droplets. Through rational design of the pattern arrangement, three-dimensional multimodal swarm behaviors of droplet array, including translation-along and rotation-around the $x$, $y$, and $z$ axis, respectively, are successfully realized. We demonstrate that plentiful and tunable swarm vibration behavior can be utilized for aseismic energy management, directional transportation, and laser modulation. Here only sinusoidal waves are used to drive the vibration of the stage, while other forms of wave, especially asymmetric waves such as the exponential wave, can lead to more diverse and interesting droplet behaviors (Supplementary Fig. 13). In addition, the liquid is not limited to water and glycol used here. Other functional liquids can also be driven in a programable way and achieve abundant applications, such as vibrating ionic liquid droplets to control the on/off state of a circuit, vibrating magnetic liquid droplets to perform as magnetic robots for complex tasks, etc. Our study provides new insights into the innovative design and facile fabrication of multimodal mechanical transducers, which would surely boost diverse applications in fields such as energy, MEMS, and optics.

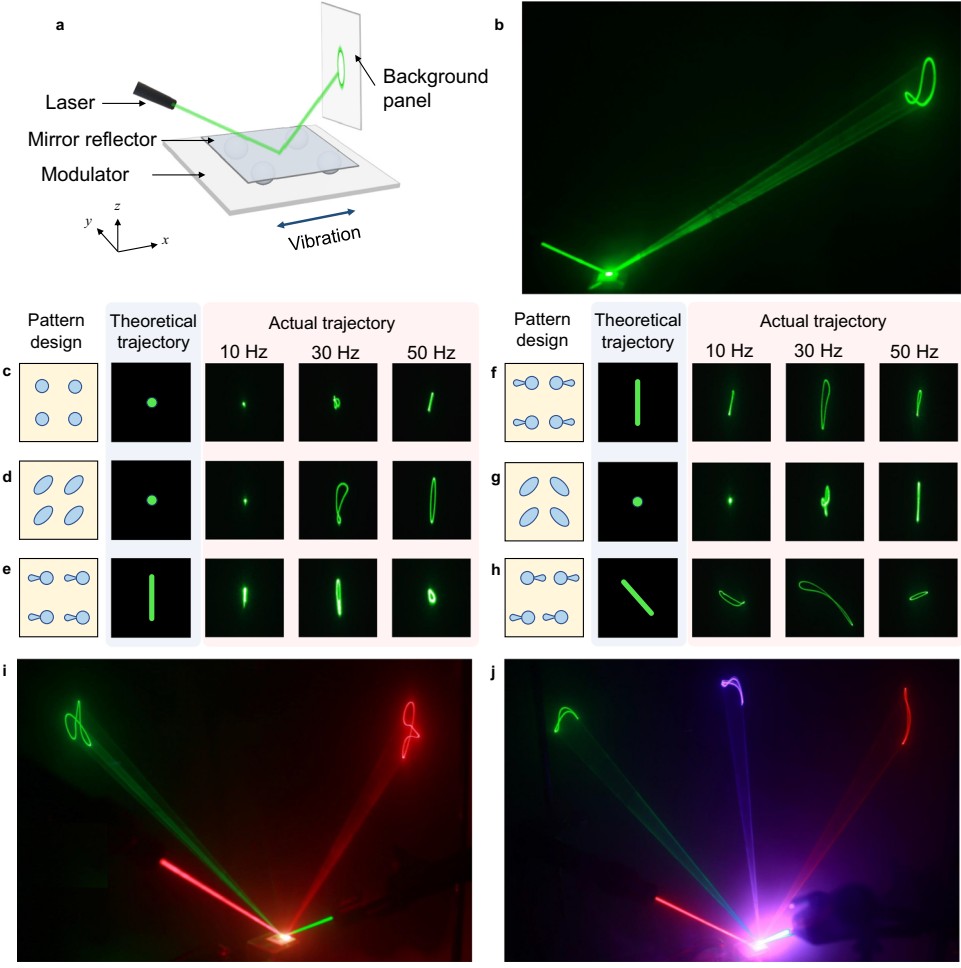

**Fig. 5 | Laser modulation using the droplet-based mechanical transducers.**
**a** The experimental setup of the laser modulation test includes a commercial laser serving as the laser source, an upper plate serving as the mirror reflector, a droplet-based mechanical transducer serving as the modulator, and a background panel. Here the top surface of the upper plate is sputtered with a high-reflective platinum layer to enhance its reflectivity. **b** The steering of one laser beam driven by the vibrating upper plate. **c–h** The theoretical and actual trajectories of laser spots modulated by the transducer systems with (**c**) circle array, (**d**) ellipse array, (**e**) gourd array, (**f**) M-gourd array, (**g**) M-ellipse array, and (**h**) R-gourd array, respectively. **i**, **j** The simultaneous steering of (**i**) two and (**j**) three laser beams by the droplet-based mechanical transducer.

## Methods

### Fabrication of patterned substrate
Beck etchant was prepared by mixing 40 mL HCl (37 wt. %), 12.5 mL deionized water and 2.5 mL HF (40 wt. %). Thin aluminum plates were washed with deionized water and then immersed in Beck etchant for 15 s at 25 °C. After corrosion, the plates were sequentially washed with deionized water and ethanol, blow-dried with nitrogen and then surface-modified with 1H, 1H, 2H, 2H-perfluorodecyltrimethoxysilane by chemical vapor deposition at 110 °C for 5 h. The obtained aluminum plates were superhydrophobic. The plates were ablated by laser (SCANcube III 10, LAJAMIN) to fabricate wettability patterns. The ablated regions became superhydrophilic.

### Fabrication of hydrogel bead
15 g acrylamide, 0.3 g N, N′- methylenebisacrylamide and 0.3 g ammonium persulfate were added into 150 mL distilled water. Then, 300 μL N, N, N′, N′- tetramethylethylenediamine was added to the solution at room temperature. After quick and thorough mixing, the solution was immediately injected into the Polydimethylsiloxane (PDMS) template. After 10 min standing, the Polyacrylamide (PAAM) hydrogel bead was synthesized by radical polymerization.

### Vibration transformation test of steel bead, hydrogel bead, and droplet
The vibration transformation test was done on a signal generator (FY6900-20M, FeelElec), a power amplifier (FPA101A, FeelElec) and a vibration excitor (SA-JZ002, Shiao Technology Co.). The frequency was 30 Hz and the amplitude is 350 μm. The superhydrophobic substrate with a superhydrophilic gourd array pattern was fixed on the vibration stage of the excitor, then the steel bead (diameter = 4 mm), the hydrogel bead (diameter = 4 mm), and a 10 μL glycol droplet were adhered to/placed on the substrate. The output vibration was recorded by a high-speed camera (Phantom V12.1, Vision Research Inc.) at 2000 frames per second from the side view.

### Influencing factors of the output symmetry of droplet
To reveal the influence of pattern design on the symmetry of output vibration, we fabricated different patterns on the substrate to pin the droplet and tune its behavior. A circle pattern (diameter = 3 mm), a dumbbell pattern (two equal-sized circles with diameter of 3 mm, connected by a rectangle channel) and a gourd pattern (two unequal-sized circles ($W_1$ = 1.5 mm, $W_2$ = 3 mm) connected by a trapezoid channel) were fabricated on the substrate. Furthermore, to investigate

the influence of gourd pattern size on the output symmetry of droplet, we fixed the diameter of the large circle to be 3 mm, and varied the diameter of the small droplet to be 0.5 mm, 1.0 mm, 1.5 mm, 2.0 mm and 2.5 mm, respectively. To investigate the influence of vibration factors, we first fixed the amplitude to be 350 μm and varied the frequency from 10 Hz to 100 Hz, and then fixed the frequency to be 30 Hz and varied the amplitude from 50 μm to 500 μm. The asymmetric coefficient was recorded.

### Three-dimensional, multi-modal swarm vibration test

Utilizing the synergistic effect of droplet array, four droplets with in-phase or anti-phase motion can be realized and used to drive the upper plate for three-dimensional, multimodal vibration behavior. We designed six different patterns, including three translational symmetric arrayed patterns (the circle array, the ellipse array, and the gourd array) and three translational asymmetric arrayed patterns (the R-gourd array, the M-gourd array, and the M-ellipse array). First, arrayed patterns were fabricated on the substrate, and four glycol droplets were pinned on the patterns. The upper plate was aluminum sheet with thickness of 0.1 mm, which is superhydrophobic with four superhydrophilic dots to pin the droplet. When the stage was vibrating with frequency of 30 Hz and amplitude of 350 μm, the vibration mode of upper plate was recorded by a high-speed camera (Phantom V12.1, Vision Research Inc.) at 2000 frames per second from the front view, the side view or the top view.

### Continuous rotation test

We designed two different arrayed patterns to realize continuous rotation of the upper plate both in clockwise and anti-clockwise direction. We fabricated four gourd patterns arranged in a clockwise/anti-clockwise circle queue on the substrate, and fabricated a superhydrophilic ring with a width of 0.5 mm on the circular superhydrophobic upper plate. The rotation behavior of the upper plate was recorded by a camera.

### Aseismic vibration management test

We built three different transducer systems to compare their aseismic vibration management ability. As a control group, the architecture composed of three building blocks was directly stacked on the bare substrate. In the other two systems, the circle-arrayed pattern and gourd-arrayed pattern were fabricated on the substrate, four 10 μL glycol droplets were placed on top, and the upper plate was placed on the droplets. Then the three building blocks were stacked on the upper plate. As shown in Supplementary Video 5, the frequency was 30 Hz and the amplitude increased from 0 to 500 μm, by 50 μm each time. To precisely quantify the vibration intensity that the three systems can bear, we set the vibration frequency to be 30 Hz, and the actuation voltage increased gradually by 0.01 V each time (corresponding to a 4 μm increase in amplitude), until the architecture collapsed. The critical amplitude of collapse was recorded. Every test was repeated five times.

### Directional object transportation test

A circle-gourd array pattern and a M-gourd array pattern were fabricated on the substrate, respectively. Droplets with volume of 15 μL were placed on each pattern. A rectangle upper plate was placed on the droplet array with its length along the vibration direction. Then an object was placed on the center region of the upper plate. When the vibration was on, the movement of the object was recorded by a camera to reveal the different transportation ability of the mechanical transducers.

### Laser modulation test

We used ultrathin glass slide (thickness: 0.05 mm) as the upper plate. The bottom side of the glass slide was modified to be superhydrophobic using a nano SiO$_2$ spray, and superhydrophilic dots were fabricated by laser to fix the droplets. A platinum layer with a thickness of 50 nm was sputtered on the top side of the glass slide to form a high-reflective mirror layer. We formed a three-layered mechanical transducer: on the wettability patterned stage, four 10 μL glycol droplets were placed, on which the upper plate was placed. When the vibration was on, we shed a laser beam onto the vibrating upper plate. Thus, the commercial laser pointer served as a laser source, the upper plate served as a mirror reflector, and the substrate-droplet system served as a modulator. The reflected laser beam moved in different paths according to the vibration mode of the upper plate, thus forming different laser trajectories on the background panel. We modulated the vibration frequency and the wettability pattern, and recorded the changes of the laser trajectory. Furthermore, we generated smoke in a dark environment to show the path of laser beams. Two/three laser pointers were used and shed laser beams on the upper plate from different directions to demonstrate the ability of the mechanical transducers to simultaneously modulate multiple laser beams.

### Reporting summary

Further information on research design is available in the Nature Portfolio Reporting Summary linked to this article.

## Data availability

The authors declare that the data supporting the findings of this study are available within the main article and the Supplementary Information. Source data are provided with this paper.

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

## Acknowledgements

We thank Prof. Claas Willem Visser for the discussion. This work is supported by the National Key R&D Program of China (Grant No. 2023YFE0111500, SQ2023YFE0101537), the National Natural Science Foundation of China (Grant No. 52321006, T2394480, T2394484, 22272182, 52293473, 51903240), Xiangfu Lab Research Project (XF012023C0100), Beijing National Laboratory for Molecular Sciences (BNLMS-CXXM-202005), the Youth Innovation Promotion Association of CAS (2023039), the China Postdoctoral Science Foundation (2023M733554), and the Beijing Nova Program (20230484291).

## Author contributions

Y.S., H.L., and L.X. conceived the idea. L.X. and A.L. performed the experiments. L.X., A.L., H.L., and Y.S. wrote the manuscript. X.Y. and K.L. helped with the figures. R.Y., X.D., R.L. and Q.L. discussed the results and helped modify the manuscript. L.X. and A.L. contributed equally.

## Competing interests

The authors declare no competing interests.
