## [Peer Review File · Nature Communications]

REVIEWER COMMENTS

Reviewer #1 (Remarks to the Author):

Review Report

For

Droplet-based reciprocity-adjustable mechanical transducers modulated by the symmetry of wettability patterns

NCOMMS-23-45848-T

Various methods have been demonstrated to achieve nonreciprocity in physical systems, where the system's response violates the fundamental reciprocal laws, rendering systems that can break time-reversal symmetry and can deviate from thermodynamic static equilibrium. Among these methods, there are passive, time-invariant systems that can also break reciprocity. While the focus has been given on enabling nonreciprocity in fields such as electromagnetics, optics, acoustics, and quantum systems, there has also been a growing interest in enabling nonreciprocity in mechanical systems.

This manuscript presents a new direction of enabling mechanical nonreciprocity by utilizing the asymmetric deformation of liquid droplets with geometric asymmetries. This casts new possibilities for developing mechanical transducers capable of converting a single-mode oscillation into multiple different modes. The method presented involves the design of superhydrophilic patterns on superhydrophobic solid surfaces, allowing droplets to adhere and take on the shape of these patterns. This approach was used to create various mechanical systems that act as mechanical transducers. These systems utilize arrays of droplets with different shapes confined between two plates, enabling mechanical transduction between a horizontally oscillating plate and a free plate. By enabling nonreciprocity in these systems, the results demonstrate their application in seismic energy control, object transportation, and laser beam modulation.

While this work has the potential to be published in Nature Communications, the authors still need to address two main aspects. First, it is essential to clearly demonstrate that the functionalities achieved utilizing the proposed mechanical systems arise solely from nonreciprocity. Second, the authors may need to establish a concrete correlation between the mechanical transduction processes and the nonreciprocity of the system.

Here are the detailed comments that authors may need to consider and address in a revised version of their work.

Comment #1 – Additional discussion surrounding the modal analysis provided in Fig. 3 may be needed. In the case of the array with circular droplets in Fig. 3c, it is important to note that this array could induce not only axial translations but also transverse translational motion as the base oscillates. This is due to the fact that the height of the circular droplet can change as the base undergoes one period of oscillation, as shown in the results in Fig. 2a, Extended Data Fig. 3, and supplemental video 2, where the height h changes with D . This height variation can result in a displacement of the top plate in the z -direction.

A similar observation can be made for the array with elliptic droplets in Fig. 3d, which could also induce a displacement in the z -direction, and probably other modes. While it is true that the translational motion in the z -direction for these arrays might be relatively small compared to the array with the gourd droplets, it is essential to emphasize how the mechanical motion conversion to a transverse motion is related to nonreciprocity. An additional sub-figure could be included to compare the transverse motion developed by the different arrays, emphasizing that this motion is negligible for patterns other than the gourd pattern. Furthermore, additional sub-figures/figures might be helpful to demonstrate the significance of the developed modes for the various patterns and arrays. Moreover, it would be important to include a discussion on the potential effect of the weight of the top plate in suppressing the transverse motion.

Comment #2 – The results for laser manipulation presented in this study are indeed intriguing. However, it is essential to note that these results seem to suggest that the modal analyses presented in Fig. 3 may be primarily applicable at low frequencies of the base vibration. Specifically, the results for laser trajectories obtained for various droplet arrays are consistent with the modes explained in Fig. 3 when the frequency is lower than 10 Hz. Nonetheless, the results obtained at higher frequencies indicate that additional modes have been developed in each array, resulting in different laser trajectories.

For example, in the case of the array with circular droplets, the plate is expected to only oscillate horizontally as shown in Fig. 3c, resulting in a laser beam that is reflected into one point at the plate center as it oscillates. This behavior was obtained only at frequencies up to 10 Hz, as shown in Fig. 5. However, at higher frequencies, other laser trajectories were obtained, indicating that the plate exhibits additional modes of vibration at these high frequencies.

The authors need to explain the reasons behind this behavior. The inertia of the plate could be an important factor that may need to be considered to explain this behavior.

Comment #3 – In page 2 and Fig.1c, it was stated that the steel bead exhibits deformation reciprocity. However, the steel bead is utilized to indicate a rigid bead, and thus it should

“hypothetically” exhibit no deformation, as shown in Fig. 1c. Thus, the term deformation reciprocity may not be consistent with this context.

Additionally, the steel bead was considered to represent the case where the bead is rigid and where neither the induced vibration nor the hydrophilic pattern produces any deformation in it. On the other side, the liquid droplet was considered to represent the case where the bead is highly elastic and can be deformed due to both the vibration and the hydrophilic pattern, resulting in deformation nonreciprocity. Taking this into account, it is reasonable to infer that the hydrogel bead used in the results in Figs. 1b and 1c can also exhibit deformation nonreciprocity. Even if it may be extremely small nonreciprocity, the hydrogel bead may represent a case between the two extreme cases for the steel and the liquid beads. Therefore, the authors may need to further clarify this in the manuscript.

Additional Comments:

1. The authors did not mention or show information about the amplitude at which the structure on the gourd array collapsed in the seismic test. This information may be important to mention and to be shown in the supplemental video.
2. I am not sure whether the term “reciprocity-adjustable” used in the title and through the paper is an accurate term to use. Reciprocity is a fundamental property of physical systems, which may not be adjusted unless the system turns to be nonreciprocal. Nonreciprocity, in contrast, can be adjusted and controlled. Therefore, the title and the text may need to be revised to better reflect “nonreciprocity adjustment” not “reciprocity adjustment”.
3. The authors may need to be more specific with some terms such as “vibration absorption”, as I don’t think that the developed system contains sources of energy dissipation. However, through the vibration manipulation, the seismic effect on the structure can be reduced.
4. It is important to emphasize that the liquid flow inside the droplet is reciprocal, while the collective deformation of the droplet can be nonreciprocal, depending on the shape of the droplet. In addition, it is important to emphasize that nonreciprocity is attained only for droplets with non-symmetric geometries.
5. It is important to accurately define the droplet height h , which is supposed to be the ultimate height attained by the droplet, at the ultimate end of the oscillation span. This also needs to be indicated in the figures, particularly in Fig. 1, using horizontal lines indicating the ultimate height attained by the droplet during the oscillation.
6. In Fig. 2, the nonreciprocity parameter η has been demonstrated depending on the geometry of the droplet and the amplitude of the base oscillation. I wonder if the authors can provide some mathematical relations linking how the nonreciprocity parameter η depends on these factors.
7. In Page 6, the authors may need to emphasize that the vibration of the stage is only along x-axis.

8. The continuous rotation example demonstrated in Extended Data Fig. 6 and in Supplemental Materials Sec. 3.3 is interesting. However, I wonder if the same behavior can be obtained using other patterns, such as the ellipse pattern. Additionally, the authors may need to emphasize how this behavior is contingent on nonreciprocity.
9. The authors may need to investigate whether the elliptic pattern of a single droplet can produce nonreciprocity like the gourd pattern, even if a relatively small nonreciprocal effect.
10. The term “showing enhanced vibration ability (Fig. 4)” is confusing, as it is anticipated to discuss the structure stability in the part.
11. In the seismic example provided, a quantitative analysis for the vibration manipulation and the seismic resistance/stability may be needed.
12. I suggest using “laser source” instead of “power source”.

Reviewer #2 (Remarks to the Author):

In this work, Li and Song et al. reported the deformation behavior of droplets placed on wettability patterned substrates when to sinusoidal vibration in one horizontal direction. A droplet placed on a circular, elliptical, or gourd-shaped superhydrophobic/superhydrophilic pattern behaves as an elastic body, where the reciprocity of its deformation under vibration can be rationally tuned by the symmetry of the pattern shape. The authors also study the swarm behavior of the reciprocal/nonreciprocal vibrating droplets. When four reciprocally or nonreciprocally designed droplets are placed in a translational-symmetrical or translational-asymmetrical geometry and vibrated simultaneously from the bottom substrate, the plate placed on top of the droplets undergoes various motions, including horizontal and vertical torsional motions. This system can be used as a vibration transducer system to modulate object transport and laser direction.

Although there have been vigorous attempts to control droplet transport under vibration by substrate patterning, the use of droplets as vibration transducers is new. Furthermore, systems with mechanical nonreciprocity have been a hot topic in materials science in recent years, but most of them have to use complex metamaterials and robots, except one example based on composite gels. The methodology proposed in this work, using droplets on a wettability patterned substrate, is completely different from conventional methodologies, and we see a lot of potential. High quality

proof-of-concept experiments to demonstrate the practicality of this system are also highly appreciated. For this reason, I strongly recommend the publication of this work in this journal.

Some comments hereafters:

(1) In Fig. 2d-f, the effects of various parameters (shape of superhydrophobic/superhydrophilic pattern, frequency of vibration, and amplitude of vibration) on the deformation nonreciprocity of droplets are investigated. The possible mechanism is described only for its dependence on pattern shape, but an explanation on frequency and amplitude dependence would be useful to the readers.

(2) In the present work, only the simplest vibration (sinusoidal vibration in one horizontal direction) is used as the driving source of the system. It would be better if a description is also given regarding the availability of more complex vibrations (biaxial vibrations, random vibrations, etc.).

(3) One of the advantages of this methodology is that a variety of functional liquids (magnetic fluids, ionic liquids, etc.) can be used as droplet sources, not only glycol and water. Of course, additional experiments using droplet sources other than glycol are clearly beyond the scope of this work, but a description of future possibilities based on the diversity of droplet sources would add to the appeal of this work.

(4) Section title "3.2 The mechanism for continuous rotation" in Supplementary Materials page 7 should be revised to "3.3 The mechanism for continuous rotation".

Reviewer #3 (Remarks to the Author):

The paper contributes a peculiar actuation mechanism based on the asymmetrical vibrations of droplets on a hydrophobic substrate and stabilized by hydrophilic patches. I found the setup to be interesting; it could be of interest to the broad readership of Nature Com.

My main concern is semantic: the authors misunderstand and misuse the notion of "reciprocity/nonreciprocity" and give it a meaning very different from the usual one found in physics or mechanics literature. I think the authors mean asymmetry not non-reciprocity. (non-reciprocity has a clear, technical, meaning; check any book on wave motion/wave mechanics). This is potentially an easy fix, except it requires the authors to re-write abstract and re-do their review of the state of the art.

The paper is weak theoretically. The demonstrations are appealing nonetheless.

Reviewer #1 (Remarks to the Author):

Comment 1: Various methods have been demonstrated to achieve nonreciprocity in physical systems, where the system's response violates the fundamental reciprocal laws, rendering systems that can break time-reversal symmetry and can deviate from thermodynamic static equilibrium. Among these methods, there are passive, time-invariant systems that can also break reciprocity. While the focus has been given on enabling nonreciprocity in fields such as electromagnetics, optics, acoustics, and quantum systems, there has also been a growing interest in enabling nonreciprocity in mechanical systems.

This manuscript presents a new direction of enabling mechanical nonreciprocity by utilizing the asymmetric deformation of liquid droplets with geometric asymmetries. This casts new possibilities for developing mechanical transducers capable of converting a single-mode oscillation into multiple different modes. The method presented involves the design of superhydrophilic patterns on superhydrophobic solid surfaces, allowing droplets to adhere and take on the shape of these patterns. This approach was used to create various mechanical systems that act as mechanical transducers. These systems utilize arrays of droplets with different shapes confined between two plates, enabling mechanical transduction between a horizontally oscillating plate and a free plate. By enabling nonreciprocity in these systems, the results demonstrate their application in seismic energy control, object transportation, and laser beam modulation.

While this work has the potential to be published in Nature Communications, the authors still need to address two main aspects. First, it is essential to clearly demonstrate that the functionalities achieved utilizing the proposed mechanical systems arise solely from nonreciprocity. Second, the authors may need to establish a concrete correlation between the mechanical transduction processes and the nonreciprocity of the system.

Reply 1: We deeply thank the reviewer for the time and effort in thoroughly reading our manuscript. Especially, we are very appreciative of the reviewer's interest in our work, as well as providing constructive suggestions, which will definitely improve our work. Encouraged and motivated by these comments, we have carefully modified the manuscript to address the relationship between the nonreciprocity, the mechanical transduction process, and the functionalities.

As the translation along and rotation around x , y , z -axis are the six basic motions in a three-dimensional world and can be integrated to achieve more complex motions, we attempt to achieve these mechanical motions using the droplet-based systems. If only reciprocal systems are used, such as the systems with circle array and ellipse array, then the droplets can only vibrate in xy -plane (translation along x -axis, translation along y -axis, rotation around z -axis), while the vibration with height variation (translation along z -axis, the tilting around x or y axis) cannot be achieved. Only if the droplets with mechanical nonreciprocity are involved, the upper plate can achieve the translation along z -axis and the tilting around x or y -axis. Thus, the correlation between the mechanical transduction processes and the mechanical reciprocity/nonreciprocity of droplets is: (1) When we combine reciprocally vibrating droplets with translational symmetric pattern arrangement, the droplet array generates in-phase vibration along x or y -axis, thus the upper plate translates along x or y -axis. (2) When we combine nonreciprocally vibrating droplets with translational symmetric pattern arrangement, the droplet array generates in-phase vibration along z -axis, thus the upper plate translates along z -axis. (3) When we combine reciprocally vibrating droplets with translational

asymmetric pattern arrangement, the droplet array generates out-of-phase vibration in xy -plane, and the upper plate rotates around z -axis. (4) When we combine nonreciprocally vibrating droplets with translational asymmetric pattern arrangement, the droplet array generates out-of-phase vibration with height variation, and the upper plate rotates around x or y -axis.

A similar logic applies in Fig. 4 and Fig. 5 to achieve the functionalities arising from the mechanical nonreciprocity of droplets. To enhance the aseismic capability of the system, we design a gourd array to induce the nonreciprocal vibration of the droplets array and drive the upper plate to vibrate along z -axis and suppress its vibration along x -axis. To transport objects directionally, we combine circle patterns and gourd patterns in one stage. Thus, the droplets on one side vibrate reciprocally, while the droplets on the other side vibrate nonreciprocally, so that asymmetric tilting can be achieved for object transport. For laser modulation, if the transducer is solely composed of reciprocal droplets, only translation and rotation in the xy -plane can be achieved, the theoretical laser trajectory is always a point. Only when nonreciprocal droplets and the associated height variation are involved, translation along z -axis, rotation around x or y -axis can be realized, so that the laser can generate more diverse trajectories.

Modifications in the manuscript:

Page 6, lines 1-4: we have added the discussion “As the translation along and rotation...” in the revised manuscript to elucidate the correlation between the mechanical nonreciprocity of droplets, the mechanical transduction processes, and the applications and functionalities.

Comment 2: Additional discussion surrounding the modal analysis provided in Fig. 3 may be needed. In the case of the array with circular droplets in Fig. 3c, it is important to note that this array could induce not only axial translations but also transverse translational motion as the base oscillates. This is due to the fact that the height of the circular droplet can change as the base undergoes one period of oscillation, as shown in the results in Fig. 2a, Extended Data Fig. 3, and supplemental video 2, where the height h changes with D . This height variation can result in a displacement of the top plate in the z -direction.

A similar observation can be made for the array with elliptic droplets in Fig. 3d, which could also induce a displacement in the z -direction, and probably other modes. While it is true that the translational motion in the z -direction for these arrays might be relatively small compared to the array with the gourd droplets, it is essential to emphasize how the mechanical motion conversion to a transverse motion is related to nonreciprocity. An additional sub-figure could be included to compare the transverse motion developed by the different arrays, emphasizing that this motion is negligible for patterns other than the gourd pattern. Furthermore, additional sub-figures/figures might be helpful to demonstrate the significance of the developed modes for the various patterns and arrays. Moreover, it would be important to include a discussion on the potential effect of the weight of the top plate in suppressing the transverse motion.

Reply 2: We thank the reviewer for the comments. We agree with the reviewer that the droplets on the circular array and the elliptical array also exhibit height variation in addition to their main movement, as demonstrated in Fig. 2a. However, the translations along z -axis are much smaller compared with the translation along x -axis for the circle array and the translation along y -axis for the ellipse array, as shown in Fig. R1. As a result, we mainly focus on the translation along x -axis for circle array and the translation along y -axis for ellipse array, and therefore ignore the translation

along z -axis for these two patterns. We have added a related discussion in the revised manuscript.

Fig. R1 The vibration amplitude along x , y , z -axis for circle array, ellipse array and gourd array.

Using pattern arrays to adjust the deformation of droplets, six different vibration modes can be achieved, including the translation along and rotation around the x , y , and z -axis. These six different modes are the basic motion modes in a three-dimensional system, and can be combined to achieve more complex motions. For example, as the droplet on the circle pattern exhibits symmetric deformation, while the droplet on the gourd pattern exhibits asymmetric deformation, combining the circle and gourd patterns on a vibrating stage can make the droplets deform symmetrically on one side and asymmetrically on the other side, thus, the upper plate can tilt unidirectionally and make the object placed on it to be transported unidirectionally.

Fig. R2 The unidirectional tilting is achieved by combining circle pattern and gourd pattern, and can be utilized in object directional transport.

Furthermore, as pointed out by the reviewer, the weight of the upper plate does have the potential to suppress the transverse motion. In our previous experiments, we used an aluminum plate as thin as possible (14.7 mg, as indicated by plate 1 in Fig. R3) and assumed that the weight of the upper plate was negligible, so that we could have a simplified modal to deduce the behavior of droplet array from the behavior of single droplet. Here, to explore the potential effect of the weight of the top plate, we compare the vibration amplitude along z -axis of transducer systems with upper plates of different weights. The weights of the plates are 14.7 mg, 40.8 mg, 134.0 mg and 454.0 mg, respectively. The vibration frequency is 30 Hz and the vibration amplitude of the stage is set as 350 μm . It's shown in Fig. R3 that when the weight of the upper plate increases, the vertical vibration

amplitude of the circle array shows a descending trend, as it decreases from 128 μm without upper plate to 22 μm with Plate 4 (454.0 mg). Further increasing the plate weight will cause coalescence or collapse of the droplets. However, when the weight increases, the translation along z -axis on the gourd array also decreases dramatically, and the translation is almost negligible when the weight of the upper plate increases to 454.0 mg, as the droplet is compressed to a pancake shape, and no inner flow appears between the large circle and the small circle. So, using a heavier upper plate will suppress the z -axis motion for the circle array, yet a lighter upper plate is preferred to show the z -axis motion for the gourd array. We have added the related discussion in the revised manuscript.

Fig. R3 The influence of the upper plate weight on its vertical vibration amplitude.

Modifications in the manuscript:

1. Page 6, lines 30-34: the sentences “The upper plate also has slight...” were added.
2. Page 6, lines 39-41: the sentences “compounded with the vibration along...” were added.
3. Page 7, lines 1-5: the sentences “Due to the x -axis...” were added.
4. Fig. R1 and Fig. R3 were added to the revised Supplementary Materials as Supplementary Fig. 8 and Fig. 9.

Comment 3: The results for laser manipulation presented in this study are indeed intriguing. However, it is essential to note that these results seem to suggest that the modal analyses presented in Fig. 3 may be primarily applicable at low frequencies of the base vibration. Specifically, the results for laser trajectories obtained for various droplet arrays are consistent with the modes explained in Fig. 3 when the frequency is lower than 10 Hz. Nonetheless, the results obtained at higher frequencies indicate that additional modes have been developed in each array, resulting in different laser trajectories.

For example, in the case of the array with circular droplets, the plate is expected to only oscillate horizontally as shown in Fig. 3c, resulting in a laser beam that is reflected into one point at the plate center as it oscillates. This behavior was obtained only at frequencies up to 10 Hz, as shown in Fig. 5. However, at higher frequencies, other laser trajectories were obtained, indicating that the plate exhibits additional modes of vibration at these high frequencies. The authors need to explain the

reasons behind this behavior. The inertia of the plate could be an important factor that may need to be considered to explain this behavior.

Reply 3: We thank the reviewers for the valuable suggestions and positive evaluation of our demonstration. As mentioned in Reply 2, the motion of the upper plate is actually a complex compounded motion, but we mainly focus on its main motion (for example, the translation along x -axis for the circle array) and ignore its vibration in other directions. Assuming that the motion of the upper plate is pure, we can deduce the theoretical laser trajectories of these six patterns, as shown in Fig. R4. For the circle array (Fig. R4c), ellipse array (Fig. R4d), and the M-ellipse array (Fig. R4g), the upper plate translates or rotates in the xy -plane, so the reflected laser trajectory is a static point. For the gourd array (Fig. R4e) and the M-gourd array (Fig. R4f), the upper plate translates along z -axis or rotates around y -axis, so the reflected laser trajectory is a vertical line. For the R-gourd array (Fig. R4h), the upper plate rotates around x -axis, so the reflected laser trajectory should be an oblique line with both x -direction component and y -direction component. In general, the actual laser trajectories of the laser match the theoretical results, except that some trajectories are assumed to be a point but are actually elongated to be a line. The reason is that the main axial translation/rotation is compounded with other forms of vibrations, so the laser trajectories may have different shapes. To be mentioned, the laser trajectory is especially sensitive to the tilting of the upper plate. Tilting of an angle less than 2° can cause the trajectory to be stretched into a line.

The influence of frequency can be qualitatively analyzed by a simple force analysis. The droplet-plate system is subjected to three forces: the inertia force, the capillary force and the viscous force. The inertia force is scaled as $(m_{\text{plate}}+m_{\text{droplet}})D_{\text{max}}f^2$, where m_{plate} , m_{droplet} , D_{max} , and f are the mass of plate, the mass of droplet, the displacement of the stage, and the vibration frequency, respectively. The inertia force tends to keep the droplets still. The capillary force is related to the deformation of droplet and will be a function of surface energy γ , droplet radius r , and the deformation of droplet which is associated with the stage displacement D_{max} and the vibration frequency f . The capillary force tends to keep the droplets in a spherical-like shape and have the minimum surface area. The viscous force is proportional to the viscosity and the velocity gradient in the droplet, and is much smaller than the inertia force and the capillary force at a frequency below 100 Hz. Take the circle array as an example, at a low frequency (~ 10 Hz), the capillary force dominates to maintain the spherical shape of the droplet, so the upper plate vibrates along x -axis in a relatively stable way, and the laser trajectories are well consist with the theoretical result. When the frequency increases to a range from 20 Hz to 60 Hz, the upper plate translates with a larger amplitude and is compounded with tilting of a larger angle, so the trajectories are vertically elongated. When the frequency increases further, due to the domination of the inertia force, the vibration amplitude of the upper plate decreases, and the sizes of the laser trajectories decrease and finally shrink to a point, as shown in Supplementary Fig. 11.

Fig. R4 The theoretical and actual laser trajectory in the laser modulation test. **a**, The experimental setup. **b**, The theoretical relation between pattern design, plate motion and laser trajectory. **c-h**, The theoretical and actual laser trajectory of **(c)** circle array, **(d)** ellipse array, **(e)** gourd array, **(f)** M-gourd array, **(g)** M-ellipse array, and **(h)** R-gourd array.

Modifications in the manuscript:

1. Page 11: the theoretical trajectories were added in Fig. 5.
2. The analysis of the influence of frequency on the laser trajectories was added in Supplementary Text 2.3.

Comment 4: In page 2 and Fig. 1c, it was stated that the steel bead exhibits deformation reciprocity. However, the steel bead is utilized to indicate a rigid bead, and thus it should “hypothetically” exhibit no deformation, as shown in Fig. 1c. Thus, the term deformation reciprocity may not be consistent with this context.

Additionally, the steel bead was considered to represent the case where the bead is rigid and where neither the induced vibration nor the hydrophilic pattern produces any deformation in it. On the other side, the liquid droplet was considered to represent the case where the bead is highly elastic and can be deformed due to both the vibration and the hydrophilic pattern, resulting in deformation nonreciprocity. Taking this into account, it is reasonable to infer that the hydrogel bead used in the results in Figs. 1b and 1c can also exhibit deformation nonreciprocity. Even if it may be extremely small nonreciprocity, the hydrogel bead may represent a case between the two extreme cases for the steel and the liquid beads. Therefore, the authors may need to further clarify this in the manuscript.

Rely 4: We thank the reviewer for the suggestions. The steel bead is totally rigid, and is not influenced by the wettability patterns, so it exhibits no deformation. On the contrary, the droplet is fluid and can be shaped by the pattern, so it deforms asymmetrically. In our work, we use a PAAM hydrogel bead with a modulus of several kPa, which makes it sufficient to maintain the shape when placed on a solid surface. When it is placed on the wettability patterned surface, it will not wet the surface and only contacts with a small area. Hence, it is reasonable to believe that the hydrogel bead is not influenced by the wettability pattern, and exhibits symmetric deformation. However, as

mentioned by the reviewer, the hydrogel bead may deform to fit the wettability pattern if it is soft enough, and an asymmetric deformation may appear.

According to the suggestions, we classify the rigid bead, the hydrogel, and the droplet into three different modes: no deformation, symmetric deformation, and asymmetric deformation. Related discussions have been updated in the revised manuscript.

Modifications in the manuscript:

1. Page 2, line 37: the phrase “no deformation” was used to describe the steal bead behavior.
2. Page 2, line 44: the phrase “deformation symmetry” was used to describe the hydrogel bead behavior.
3. Page 3, line 17: the phrase “deformation asymmetry” was used to describe the droplet behavior.
4. Page 3: Fig. 1b and 1d was updated.

Comment 5: The authors did not mention or show information about the amplitude at which the structure on the gourd array collapsed in the seismic test. This information may be important to mention and to be shown in the supplemental video.

Reply 5: We thank the reviewer for the suggestions. In the seismic test, the average amplitude that the structure on the gourd array collapses is $481 \pm 93 \mu\text{m}$, which is 1.8 times of that directly on the stage ($276 \pm 38 \mu\text{m}$) and 3 times of that using the circle array ($160 \pm 26 \mu\text{m}$), showing enhanced seismic resistance performance. We have modified the related discussion in the manuscript.

Modifications in the manuscript:

1. Page 8, lines 18-19, Page 9, line 6: the results of the collapse amplitude for the gourd array were added.
2. The video of the structure collapse was added in Supplementary Video 5.

Comment 6: I am not sure whether the term “reciprocity-adjustable” used in the title and through the paper is an accurate term to use. Reciprocity is a fundamental property of physical systems, which may not be adjusted unless the system turns to be nonreciprocal. Nonreciprocity, in contract, can be adjusted and controlled. Therefore, the title and the text may need to be revised to better reflect “nonreciprocity adjustment” not “reciprocity adjustment”.

Reply 6: We thank the reviewer for the valuable suggestions. In our previous submission, we consider “reciprocity” as a broad definition, including both reciprocal behaviors and nonreciprocal behaviors. We agree that when a system is reciprocal, it can only be broken into a nonreciprocal system, but cannot be adjusted. We have revised the title and also adjusted the expressions in the text accordingly.

Modifications in the manuscript:

1. Page 1: the title was modified to be “Droplet-based reciprocity-regulating mechanical transducers modulated by wettability patterns”.
2. Page 1, line 18 and Page 2, lines 18: the related expressions have been modified to be “regulating”.

Comment 7: The authors may need to be more specific with some terms such as “vibration absorption”, as I don’t think that the developed system contains sources of energy dissipation. However, through the vibration manipulation, the seismic effect on the structure can be reduced.

Reply 7: We thank the reviewer for the comment. When using the gourd array in the seismic test, the x -axis vibration amplitude of the upper plate is decreased compared with the stage, thus the stability of the system is improved. The reason can be attributed to two categories: on one side, part of the horizontal vibration is transformed into vibration of other forms, mainly the vibration along z -axis. On the other side, the neck in the gourd pattern can induce vigorous liquid flow between the large and small circles during the droplet vibrating, which will dissipate most of the vibration energy and reduce the seismic effect on the structure. To address the concern of the reviewer, we use “vibration management” to replace “vibration absorption”, as it can be divided into energy transformation and energy dissipation, both of which contribute to the enhancement of the aseismic capability of the system.

Modifications in the manuscript:

Page 1, line 26 and Page 2, line 28: the term “vibration absorption” was replaced by “vibration management”.

Comment 8: It is important to emphasize that the liquid flow inside the droplet is reciprocal, while the collective deformation of the droplet can be nonreciprocal, depending on the shape of the droplet. In addition, it is important to emphasize that nonreciprocity is attained only for droplets with non-symmetric geometries.

Reply 8: We thank the reviewer for the advice. It is indeed that at a microscopic level, the liquid flow inside a droplet is reciprocal, but the net flow rate, as well as the droplet deformation, is asymmetric at the macro level. As pointed out by the reviewer, the nonreciprocity of the droplet is indeed caused by the asymmetric droplet morphology or the asymmetric pattern design.

Modifications in the manuscript:

Page 4, lines 38-40: we have emphasized that the nonreciprocity originates from the asymmetric pattern design and asymmetric droplet morphology in the revised manuscript.

Comment 9: It is important to accurately define the droplet height h , which is supposed to be the ultimate height attained by the droplet, at the ultimate end of the oscillation span. This also needs to be indicated in the figures, particularly in Fig. 1, using horizontal lines indicating the ultimate height attained by the droplet during the oscillation.

Reply 9: We thank the reviewer for the comments. The droplet height h is defined as the height difference between the vertex of droplet and the stage during the vibration, and h_0 is defined as the droplet height when it is static. The droplet height fluctuates during vibration, as shown in Fig. 2a. We compare the droplet height at the leftmost position (h_{leftmost}) and the rightmost position ($h_{\text{rightmost}}$) to determine whether the droplet deformation is symmetric.

Modifications in the manuscript:

1. Page 3, lines 8-10: the sentence “the droplet height h ...” was added.
2. Page 3: a horizontal line was added in Fig. 1d to indicate the droplet height.

Comment 10: In Fig. 2, the nonreciprocity parameter η has been demonstrated depending on the geometry of the droplet and the amplitude of the base oscillation. I wonder if the authors can provide some mathematical relations linking how the nonreciprocity parameter η depends on these factors.

Reply 10: We thank the reviewer for the comment. A mediate diameter ratio of the gourd pattern can induce the largest asymmetry coefficient η , as the small circle can effectively shunt the liquid, meanwhile, the pattern is asymmetric enough to maximize the droplet height difference between the leftmost and rightmost positions. A mediate vibration amplitude is also preferred (Fig. R5b), as small vibrations cause slight deformation of the droplet, and the height variation between the leftmost and rightmost position is relatively small, resulting in a small value of η (Fig. R5a). On the contrary, large vibrations cause an overviolent flow of liquid that drives most of the liquid into the small circle (Fig. R5c), which reduces the droplet height difference between the leftmost and rightmost positions and thus weakens the effect of the patterns. However, the three-dimensional solid-liquid system with various droplet geometries may cause complex and unpredictable liquid flow or vortices, which makes it difficult to deduce a mathematical equation to relate the asymmetry coefficient η and the influencing factors. We have modified the related discussions in the revised manuscript.

Fig. R5 The height variation with different vibration amplitude. The amplitude is (a) small, (b) intermediate, and (c) large, respectively.

Modifications in the manuscript:

1. Page 4, line 42: the sentence “For gourd patterns with different...” was added.
2. Page 5, lines 9-13: the sentence “By contrast, the asymmetric coefficient...” was added.

Comment 11: In Page 6, the authors may need to emphasize that the vibration of the stage is only along x-axis.

Reply 11: We thank the reviewer for the advice. We have emphasized this point in the revised manuscript.

Modifications in the manuscript:

1. Page 6, line 7: the sentence “the stage is driven by a sinusoidal wave along x-axis” was added.
2. Page 7: the direction of vibration was pointed out using an arrow in Fig. 3a.

Comment 12: The continuous rotation example demonstrated in Extended Data Fig. 6 and in Supplemental Materials Sec. 3.3 is interesting. However, I wonder if the same behavior can be obtained using other patterns, such as the ellipse pattern. Additionally, the authors may need to emphasize how this behavior is contingent on nonreciprocity.

Reply 12: We thank the reviewer for the interest in our results. The key to continuous rotation is the

construction of the nonreciprocal patterns in order to achieve a directional liquid flow on the patterns and have a continuous liquid channel on the upper plate. In our work, we fabricate four gourd patterns, arranged in a circle queue on the stage, and fabricate a hydrophilic ring on the upper plate. When the liquid flows from the big circle to the small circle, a larger Laplace pressure is generated due to the large fluid volume of the big circle and the small fluid capability of the small circle. Thus, the liquid is beyond the bearing capability of the small circle and tends to flow through the liquid channel on the upper stage. On the contrary, when the liquid flows from the small circle to the big circle, a smaller Laplace pressure is created and the large circle can bear the liquid volume fluctuation. As a result, the upper plate obtains a directional force pointing from the big circle to the small circle, so it can rotate continuously.

In this case, the continuous rotation cannot be obtained using the ellipse pattern, as the droplet deformation is symmetric without any differentiated flow trend. However, unidirectional rotation can be achieved by some other asymmetric patterns, as shown in Fig. R6, as long as the droplet on it flows in a preferred direction. The pattern of triangles arranged in a circle queue can also generate a trend to rotate unidirectionally.

Fig. R6 The unidirectional, continuous rotation achieved by (a) the combination of circle and ellipse, (b) the combination of circle and polygon.

Modifications in the manuscript:

The mechanism of continuous rotation was added in Supplementary Text 2.2.

Comment 13: The authors may need to investigate whether the elliptical pattern of a single droplet can produce nonreciprocity like the gourd pattern, even if a relatively small nonreciprocal effect.

Reply 13: We thank the reviewer for the comment. If a pattern has two mutually perpendicular symmetry axes, the deformation of the droplet must be symmetric, with a η equal to 0. For the elliptical pattern, the existence of the two mutually perpendicular symmetry axes determines that the droplet deforms symmetrically with η of 0, even if it is rotated, as shown in Fig. R7 b-c. However, if we use a pattern without two mutually perpendicular symmetry axes, the droplet may generate asymmetric deformation. For example, when we use an ovoid pattern, which is the variant of the elliptical pattern, the droplet deformation shows asymmetry along x -axis, and the asymmetry coefficient can be tuned by the pattern parameters (1.3%, 5.1%, and 3.8% for three different ovoid

patterns, respectively, Fig. R7 d-f). However, the asymmetry effect is relatively small compared with the gourd pattern of 24%.

Fig. R7 The deformation symmetry/asymmetry produced by ellipse pattern and its variants. **a**, The design principle of the patterns. **b**, The droplet on an ellipse pattern deforms symmetrically, with an asymmetry coefficient $\eta = 0$. **c**, The droplet on a rotated ellipse pattern deforms symmetrically along x -axis, with $\eta = 0$. However, it causes a translation along y -axis. **d-f**, The droplet on the ovoid patterns deform asymmetrically, with $\eta \neq 0$. The shape parameters and the coefficients are **(d)** $D_1 = 2.5$ mm, $D_2 = 1.5$ mm, $\eta = 1.3\%$, **(e)** $D_1 = 3.0$ mm, $D_2 = 1.0$ mm, $\eta = 5.1\%$, **(f)** $D_1 = 3.5$ mm, $D_2 = 0.5$ mm, $\eta = 3.8\%$, respectively.

Comment 14: The term “showing enhanced vibration ability (Fig. 4)” is confusing, as it is anticipated to discuss the structure stability in the part.

Reply 14: We thank the reviewer for the comment. When using the circle pattern, the horizontal vibration amplitude of the upper plate is larger than the stage, so the structure on the upper plate is easier to collapse, showing reduced stability. On the contrary, when using the gourd pattern, the horizontal vibration amplitude of the upper plate is smaller than the stage, thus the structure can withstand stronger vibration and shows significantly enhanced seismic resistance performance. Considering the ambiguity that the phrase may raise, we replaced the term “enhanced vibration ability” with “reduced stability”.

Modifications in the manuscript:

1. Page 8, line 17: “enhanced vibration ability” was replaced with “reduced stability”.
2. Page 9: the “enhanced vibration ability” in Fig. 4b was replaced with “reduced aseismic stability”.

Comment 15: In the seismic example provided, a quantitative analysis for the vibration manipulation and the seismic resistance/stability may be needed.

Reply 15: We thank the reviewer for the suggestions. In the seismic test, we place the droplet arrays on the circle array pattern and the gourd array pattern, and drive the stage with a vibration of 30 Hz. When we increase the amplitude from 50 μ m to 500 μ m, we measure the vibration amplitude of the upper plate along x -axis, as the shear force caused by the horizontal vibration is the main reason for

architecture collapse. The results are shown in Fig. R8. It indicates that the plate on the circle array exhibits larger amplitude than the stage, resulting in its reduced stability, while the plate on the gourd array exhibits smaller amplitude than the stage, resulting in its enhanced stability. Thus, we can manipulate the vibration intensity using different patterns and for application in fields such as aseismic design. We have added the related discussions in the revised manuscript and Supplementary materials.

Modifications in the manuscript:

1. Page 8, line 15: the sentence “the horizontal vibration of...” was added.
2. Page 8, line 18: the sentence “the horizontal vibration of...” was added.
2. The quantitative analysis was added in Supplementary Fig. 10a.

Fig. R8 The vibration management ability of the circle array and the gourd array.

Comment 16: I suggest using “laser source” instead of “power source”.

Reply 16: We thank the reviewer for the suggestion. The term is replaced.

Modifications in the manuscript:

Page 9, lines 14-15 and Page 11, line 3: the term “power source” was replaced with “laser source”.

Finally, we want to thank the reviewer again for these thoughtful and insightful comments. The manuscript has greatly benefited from these valuable comments.

Reviewer #2 (Remarks to the Author):

Comment 1: In this work, Li and Song et al. reported the deformation behavior of droplets placed on wettability patterned substrates when to sinusoidal vibration in one horizontal direction. A droplet placed on a circular, elliptical, or gourd-shaped superhydrophobic/superhydrophilic pattern behaves as an elastic body, where the reciprocity of its deformation under vibration can be rationally tuned by the symmetry of the pattern shape. The authors also study the swarm behavior of the reciprocal/nonreciprocal vibrating droplets. When four reciprocally or nonreciprocally designed droplets are placed in a translational-symmetrical or translational-asymmetrical geometry and vibrated simultaneously from the bottom substrate, the plate placed on top of the droplets undergoes various motions, including horizontal and vertical torsional motions. This system can be used as a vibration transducer system to modulate object transport and laser direction.

Although there have been vigorous attempts to control droplet transport under vibration by substrate patterning, the use of droplets as vibration transducers is new. Furthermore, systems with mechanical nonreciprocity have been a hot topic in materials science in recent years, but most of them have to use complex metamaterials and robots, except one example based on composite gels. The methodology proposed in this work, using droplets on a wettability patterned substrate, is completely different from conventional methodologies, and we see a lot of potential. High quality proof-of-concept experiments to demonstrate the practicality of this system are also highly appreciated. For this reason, I strongly recommend the publication of this work in this journal.

Reply 1: We thank the reviewer for the time and efforts in reading our manuscript and raising positive evaluation of “the use of droplets as vibration transducers is new”, “The methodology proposed in this work, using droplets on a wettability patterned substrate, is completely different from conventional methodologies, and we see a lot of potential”, “High quality proof-of-concept experiments to demonstrate the practicality of this system are also highly appreciated”. In the following, we have addressed the concerns of the reviewer point by point.

Comment 2: In Fig. 2d-f, the effects of various parameters (shape of superhydrophobic/superhydrophilic pattern, frequency of vibration, and amplitude of vibration) on the deformation nonreciprocity of droplets are investigated. The possible mechanism is described only for its dependence on pattern shape, but an explanation on frequency and amplitude dependence would be useful to the readers.

Reply 2: We sincerely thank the reviewer for the suggestions. The asymmetric coefficient η can be modulated by vibration parameters, including frequency and amplitude. The dependence of η on the vibration frequency shows a complex trend, which reaches the peak at 30 Hz. This is probably because the droplets resonate at specific frequencies, and the wettability pattern alters the intrinsic frequency of the droplets. By contrast, the asymmetric coefficient exerts a simple and monotonically increasing relationship with the vibration amplitude, until reaching the maximum of 0.24. Further enlarging the amplitude will cause the overviolent flow of liquid, thus the effect of the pattern is weakened and its asymmetric coefficient is decreased.

Modifications in the manuscript:

Page 5, lines 5-13: an explanation of the influence of frequency and amplitude was added: “Besides the pattern design...”.

Fig. R9 The influence of (a) vibration frequency and (b) amplitude on the asymmetric coefficient ($n = 5$).

Comment 3: In the present work, only the simplest vibration (sinusoidal vibration in one horizontal direction) is used as the driving source of the system. It would be better if a description is also given regarding the availability of more complex vibrations (biaxial vibrations, random vibrations, etc.).

Reply 3: We thank the reviewer for the constructive suggestion. The utilization of more complex vibrations is helpful in improving the integrity of our work and may lead to more interesting phenomena or mechanisms. In our previous manuscript, we only adopt the sinusoidal vibration along x -axis as it is one of the simplest vibration modes, and it is symmetric so that the deformation asymmetry of the droplets can be totally attributed to the wettability patterns. According to the suggestion, we use four different vibration modes to drive the droplet, including the sinusoidal wave, the triangular wave, the positive exponential wave, and the sinusoidal sweep wave. The vibration modes are shown in Figure R10a, d, g, j.

The sinusoidal wave and the triangular wave are both symmetric, so the droplets exhibit similar behaviors. The droplets deform symmetrically on the circle pattern, exhibiting an asymmetry coefficient η of 0 (Fig. R10b, e). On the contrary, the droplets deform asymmetrically on the gourd pattern, exhibiting the highest height on the rightmost position and the lowest height on the leftmost position (Fig. R10c, f). The mechanism of the above behaviors has been discussed in the manuscript (Page 4, lines 31-38). However, when we use an asymmetric wave to drive the droplet, such as a positive exponential wave, the symmetry of the droplet deformation on the circle pattern is broken. The droplet on the circle pattern deforms asymmetrically, with its $h_{\text{leftmost}} > h_{\text{rightmost}}$ (Fig. R10h). Especially, the droplet exhibits a very interesting behavior on the gourd pattern when driven by the positive exponential wave. When we reverse the direction of the input vibration, the droplet shows totally different behaviors, with its asymmetry either enhanced or diminished (Fig. R10i). When we use a sinusoidal sweep wave to drive the droplet, the situation is rather complex, as the vibration hysteresis loop shown in Fig. R10k and R10l for several periods.

Modifications in the manuscript:

1. Page 11, lines 16-18: We have added related discussions about the availability of more complex vibrations “here only sinusoidal waves...”
2. Fig. R10 was added in Supplementary materials as Supplementary Fig. 13.

Fig. R10 Droplet behaviors driven by different vibration modes. **a-c**, The **(a)** wave modes and the corresponding droplet behavior on **(b)** circle pattern and **(c)** gourd pattern driven by a sinusoidal wave. **d-f**, The **(d)** wave modes and the corresponding droplet behavior on **(e)** circle pattern and **(f)** gourd pattern driven by a triangular wave. **g-i**, The **(g)** wave modes and the corresponding droplet behavior on **(h)** circle pattern and **(i)** gourd pattern driven by a positive exponential wave. **j-l**, The **(j)** wave modes and the corresponding droplet behavior on **(k)** circle pattern and **(l)** gourd pattern driven by a sinusoidal sweep wave.

Comment 4: One of the advantages of this methodology is that a variety of functional liquids (magnetic fluids, ionic liquids, etc.) can be used as droplet sources, not only glycol and water. Of course, additional experiments using droplet sources other than glycol are clearly beyond the scope of this work, but a description of future possibilities based on the diversity of droplet sources would add to the appeal of this work.

Reply 4: We thank the reviewer for the valuable suggestions. This methodology can be applied for manipulating a variety of liquids as long as it can be confined by the wettability patterns. In our work, we applied experiments on water and glycol droplets considering their availability and representativeness. Other functional liquids can also be used, such as magnetic fluids and ionic liquids, to achieve abundant applications. For example, when driving an ionic liquid droplet to vibrate in a programmable way, we may control the on and off state of a circuit. When driving a magnetic liquid droplet to vibrate, we can apply external magnetic field to the droplet, which may make the droplets generate more complex and interesting vibrating behaviors and function as

magnetic robots to complete complex tasks. The droplet sources that can be used is diverse, and the potential applications are promising.

Modifications in the manuscript:

Page 11, line 19: related discussion was added based on the diversity of droplet sources “In addition, the liquid is not limited...”

Comment 5: Section title "3.2 The mechanism for continuous rotation" in Supplementary Materials page 7 should be revised to "3.3 The mechanism for continuous rotation".

Reply 5: Thanks for the valuable comment. We have corrected the title and a throughout check was performed to correct the typos and inappropriate words.

Finally, we want to thank the reviewer again for these constructive suggestions. The manuscript has greatly benefited from these valuable comments.

Reviewer #3 (Remarks to the Author):

Comment: The paper contributes a peculiar actuation mechanism based on the asymmetrical vibrations of droplets on a hydrophobic substrate and stabilized by hydrophilic patches. I found the setup to be interesting; it could be of interest to the broad readership of Nature Com.

My main concern is semantic: the authors misunderstand and misuse the notion of "reciprocity/nonreciprocity" and give it a meaning very different from the usual one found in physics or mechanics literature. I think the authors mean asymmetry not non-reciprocity. (non-reciprocity has a clear, technical, meaning; check any book on wave motion/wave mechanics). This is potentially an easy fix, except it requires the authors to re-write abstract and re-do their review of the state of the art.

The paper is weak theoretically. The demonstrations are appealing nonetheless.

Reply: We sincerely thank the reviewer for the high evaluation of our work, and for raising this very important and constructive suggestion. Based on the suggestion, we have checked the definition and discussion of "reciprocity", as elaborated below.

Reciprocity is a general, fundamental principle governing various physical systems, which ensures that the transfer function between any two points in space is identical regardless of geometrical or material asymmetries¹, and codifies a relation of symmetry between action and reaction². James Clerk Maxwell is one of the first researchers to define the property of reciprocity. He defines that for four arbitrary points B, C, D, E in a continuous medium: "The extension in BC, due to unity of tension along DE, is always equal to the tension in DE due to unity of tension in BC"³. Enrico Betti extends the concept of reciprocity to the work done by static forces on an elastic body. So, the Maxwell-Betti theorem is mathematically formulated as $F_{AUB-A} = F_{BUA-B}$, in which F_A (F_B) is the applied force at point A (B) and u_{A-B} (u_{B-A}) is the displacement at point B (A) induced by F_A (F_B)⁴. In the fields of acoustics and elastodynamics, a reciprocal theorem, namely, that "the vibration excited at A will have at B the same relative amplitude and phase as if the places were exchanged," was stated and proved by Lord Rayleigh in 1871⁵.

Recently, the term "mechanical nonreciprocity" has been introduced to the material field, and is depicted as "the realization of the mechanical nonreciprocity requires breaking either the time-reversal symmetry or the material deformation symmetry"⁶. As shown in Fig. R11, in the work of Wang et al., they state "if a system responds to an input in one way, then it also does the same when the input is reversed"⁷, and the "mechanical nonreciprocity" is utilized to characterize the asymmetric deformation of materials (Science, 2023, 380, 192-198, as shown in Fig. R11).

We agree with the reviewer that the "deformation reciprocity/nonreciprocity" should be "deformation symmetry/asymmetry", and we have modified these descriptions in the revised manuscript accordingly. Meanwhile, we use the deformation asymmetry of the droplet to characterize the extent of mechanical nonreciprocity of the droplet-based system.

Fig. R11 In a recent publication, the mechanical nonreciprocity is characterized by the asymmetric deformation of the gel (Science, 2023, 380, 192-198). The nonreciprocal gel shows asymmetric deformation in response to shear forces applied at the top of the gel⁷.

Modifications in the manuscript:

1. The term “deformation reciprocity/nonreciprocity” was replaced by “deformation symmetry/asymmetry” in the manuscript (Page 2, lines 21, 25, 44; Page 3, line 17; Page 4, lines 4, 6, 11, 41-42).
2. The definition and characterization of mechanical nonreciprocity was added in Page 1, lines 34-36 and Page 4, lines 10-13.

In addition, quantitative characterization and discussions are added in the manuscript to strengthen the mechanism discussion of this work. We have added discussions about the influence of vibration parameters on the asymmetric coefficient, and quantified the vibration amplitudes of different translation modes. Furthermore, we have added force analysis and discussion about the influence of frequency on the laser modulation trajectories.

Modifications in the manuscript:

1. Page 5, lines 5-13: The discussion about the influence of vibration parameters on the asymmetric coefficient was added: “Besides the pattern design...”
2. We have quantified the vibration amplitude of different translation modes to demonstrate our choice of the main motions. For the circle array, the upper plate mainly translates along x -axis, while its movement amplitude along y and z -axis is very slight compared with x -axis (Fig. R11), so we focus on the x -axis translation as its main motion. For the ellipse array, the upper plate exhibits oblique motion compounded as the translation along x and y -axis, while its translation along z -axis is very slight and can be ignored. For the gourd array, the transducer system induces the translation along z -axis and diminishes its translation along x -axis, and its vibration along y -axis can be ignored. Although the upper plate always exhibits a compounded movement, we mainly focus on the x -axis translation for the circle array, the y -axis translation for the ellipse array, and the z -axis translation for the gourd array (Page 6, lines 30-33, 39-41; Page 7, lines 1-5 in the manuscript, Supplementary Fig. 8).
3. We have added force analysis and discussion about the influence of frequency on the laser modulation trajectories in Supplementary materials 2.3: “The influence of frequency ...”

Fig. R11 The vibration amplitude along x , y , z -axis for circle array, ellipse array and gourd array.

Finally, we want to thank the reviewer again for these thoughtful and insightful comments. The manuscript has greatly benefited from these valuable comments.

References:

- 1 Coullais, C., Sounas, D. & Alù, A. Static non-reciprocity in mechanical metamaterials. *Nature* **542**, 461-464 (2017).
- 2 Nassar, H. *et al.* Nonreciprocity in acoustic and elastic materials. *Nat. Rev. Mater.* **5**, 667-685 (2020).
- 3 Maxwell, J. C. L. On the calculation of the equilibrium and stiffness of frames. *The London, Edinburgh, and Dublin Philosophical Magazine and Journal of Science* **27**, 294-299 (1864).
- 4 Betti, E. Teoria della elasticita. *Il Nuovo Cimento (1869-1876)* **7**, 5-21 (1872).
- 5 Strutt, J. W. Some General Theorems relating to Vibrations. *Proceedings of the London Mathematical Society* **s1-4**, 357-368 (1871).
- 6 Shaat, M. Nonreciprocal elasticity and the realization of static and dynamic nonreciprocity. *Sci. Rep.* **10**, 21676 (2020).
- 7 Wang, X. *et al.* Mechanical nonreciprocity in a uniform composite material. *Science* **380**, 192-198 (2023).

Reviewers' comments:

Reviewer #1 (Remarks to the Author):

I extremely appreciate the authors' dedication to addressing the comments raised in the initial review process. The authors have diligently incorporated my suggestions and have provided supplementary results to address the raised comments and to emphasize the key concepts and findings of this study. Consequently, I strongly recommend the publication of the current version of the manuscript in Nature Communications.

Reviewer #2 (Remarks to the Author):

The authors thoroughly addressed all of my previous comments and questions properly. In particular, the experimental result responding to my comment 3 (Supplementary Figure 13) is beyond my imagination and quite interesting. Now I recommend this work for the publication.

Reviewer #3 (Remarks to the Author):

The term "non-reciprocity" remains misused and misappropriated. In their response the authors appear to agree that non-reciprocity is different from asymmetry but the text says "It clearly shows that when the stage vibrates in opposite directions, the response heights of the droplet are unequal at the two ends, indicating that the droplet shows deformation asymmetry, which is the characteristic of mechanical nonreciprocity." Thus in my opinion, claims of non-reciprocity, in the context of the mechanical system, remain unsubstantiated and factually incorrect.

I cannot recommend acceptance.

Reviewer #4 (Remarks to the Author):

The authors present a droplet-based transducer with asymmetric mechanical response. I carefully read the paper and the correspondence with the previous reviewers, and I tend to agree with the opinion of Reviewer 3 that this paper, while reporting interesting experiments, shows a poor understanding of the principle of mechanical reciprocity, and appears to oversell the reported results. It is surprising to me that the authors replied to the criticism of Reviewer 3 citing the Maxwell-Betti theorem, and referencing to the recent paper in *Science*, 2023, 380, 192-198, even copying a figure from that paper, which indeed showcases a nonreciprocal response. On the contrary, from what I can tell none of the results reported by the authors appears to demonstrate a nonreciprocal response. If the authors want to connect their results to a reciprocity-breaking device, they should indeed perform a measurement similar to the one they report from Fig. R11 in their own report, demonstrating that opposite forces applied to opposite positions of the sample lead to a different displacement. This would be a measurement of the degree of nonreciprocity of their mechanical system. It would also be good to compare the degree of broken nonreciprocity to the amplitude of the applied force, and to the recently reported systems that produce these effects, to demonstrate the effectiveness of their droplet-based setup. As it is now, the paper does not demonstrate nonreciprocal responses, hence the many references to reciprocity in the text are unwarranted. I do not recommend publication given this important deficiency.

Reviewer #1 (Remarks to the Author):

Comments: I extremely appreciate the authors' dedication to addressing the comments raised in the initial review process. The authors have diligently incorporated my suggestions and have provided supplementary results to address the raised comments and to emphasize the key concepts and findings of this study. Consequently, I strongly recommend the publication of the current version of the manuscript in Nature Communications.

Reply: We deeply thank the reviewer for the time and efforts in thoroughly reading our manuscript. We are extremely appreciative of the reviewer's constructive suggestions, which helped us greatly to improve our paper.

Reviewer #2 (Remarks to the Author):

Comments: The authors thoroughly addressed all of my previous comments and questions properly. In particular, the experimental result responding to my comment 3 (Supplementary Figure 13) is beyond my imagination and quite interesting. Now I recommend this work for the publication.

Reply: We deeply thank the reviewer for the time and efforts in thoroughly reading our manuscript. Especially, we are thankful for the comments and questions, which helped us a lot in improving the completeness and universality of our work.

Reviewer #3 (Remarks to the Author):

Comments: The term “non-reciprocity” remains misused and misappropriated. In their response the authors appear to agree that non-reciprocity is different from asymmetry but the text says “It clearly shows that when the stage vibrates in opposite directions, the response heights of the droplet are unequal at the two ends, indicating that the droplet shows deformation asymmetry, which is the characteristic of mechanical nonreciprocity.” Thus in my opinion, claims of non-reciprocity, in the context of the mechanical system, remain unsubstantiated and factually incorrect.

I cannot recommend acceptance.

Reply: We deeply thank the reviewer for the time and efforts in reading our manuscript and providing constructive suggestions. Based on the reviewer’s comments, we have carefully read the literature to clarify the definition of mechanical non-reciprocity. We find that “mechanical non-reciprocity” refers to the difference in the ease of material deformation in two opposite directions, meaning that the same force can cause different extents of deformation (Science, 380, 192-198), which is indeed different from our previous understanding. We agree with the reviewer that the term “symmetry/asymmetry” is more appropriate than “reciprocity/nonreciprocity”.

In addition, according to the reviewer’s comment “claims of non-reciprocity remain unsubstantiated”, we have added experiments to determine the non-reciprocity of the mechanical system by comparing its ease of deformation in opposite directions. Firstly, we rotate the substrate of the single-droplet system in clockwise and anticlockwise directions to observe the droplet deformations. As shown in Figure R1, when tilting the substrate with a circle pattern 90 degrees anticlockwise, the peak of the droplet displays a displacement of 0.41 mm, which is the same as the droplet on the clockwise substrate. However, when tilting a substrate with a gourd pattern, the deformations of the droplets are significantly different. Compared to the 3.82 mm displacement for the anticlockwise substrate, the droplet on the clockwise substrate displays a much smaller displacement (0.78 mm). The result suggests the existence of non-reciprocity in the droplet system.

Figure R1 Reciprocity characterization of the single-droplet system. (a) Droplet on a circle pattern exhibits mechanical reciprocity. (b) Droplet on a gourd pattern exhibits mechanical nonreciprocity.

In addition, we want to obtain the quantitative relationship between the deformation extent of the droplet and the exerted force. We use a plastic sheet (PET) as a cantilever beam to push the upper plate of the mechanical system and measure the force. The force F can be obtained by the deflection equation of the cantilever beam, which is:

$$F = \frac{Ebh^3W}{4l^3}$$

where elastic modulus $E = 2000$ MPa, beam width $b = 1$ mm, beam thickness $h = 0.07$ mm, beam length $l = 10.35$ mm, and W is the deflection of the cantilever beam.

As shown in Figure R2a-b, the mechanical system with circle-arrayed pattern shows mechanical reciprocity because the same shear force induces the same displacement when pushing the upper plate to move leftwards or rightwards. However, for the mechanical system with gourd-arrayed pattern, a force of 0.4 mN can move the upper plate leftwards with a displacement of 1.05 mm, but only 0.65 mm rightward displacement is achieved (Figure R2c-d). The differentiated ease of deformation in opposite directions arises from the asymmetry of the gourd pattern and the droplet morphology, exhibiting the mechanical non-reciprocity of a mechanical system with the gourd-arrayed pattern.

Figure R2 Mechanical reciprocity characterization of the droplet-arrayed mechanical system. (a-b) For the mechanical transducer with a circle array, the same shear force induces the same displacement to the left or right, showing mechanical reciprocity. (c-d) For the mechanical transducer with a gourd array, the same shear force induces different displacements to the left or right, showing mechanical nonreciprocity.

After careful consideration of the reviewers' comments, we agree that asymmetry is more

appropriate than non-reciprocity in our study, even though the system also exhibits mechanically non-reciprocal behavior. We have revised the manuscript thoroughly and used the term “transformation symmetry/asymmetry” instead of “mechanical reciprocity/nonreciprocity” to describe the ability to transform a symmetric input into symmetric/asymmetric outputs. The data about mechanical reciprocity is added in supplementary materials. We are very grateful to the reviewer for the persistence in pointing out the problem, which has greatly benefited us.

Modifications in the manuscript:

1. The title of the manuscript is revised as “Droplet-based mechanical transducers modulated by the symmetry of wettability patterns”.
2. Page 1, Lines 31-37: the sentences “From pistons driving gears ...” are added to illustrate the importance of mechanical energy conversion.
3. Page 2, Lines 43-44; Page 3, Lines 16-17; Page 4, Lines 19-20, 24, 35: the term “symmetric/asymmetric output” is used to describe the property of output vibration.
4. Page 1, Lines 12, 21; Page 4, Lines 8, 10, 39-40; Page 5, Line 15: the term “transformation symmetry/asymmetry” is used to describe the ability of the mechanical transducer to transform a symmetric input into symmetric/asymmetric outputs.
5. Page 4, Lines 35-38: The measurement result of the droplet reciprocity/nonreciprocity are added.

Modifications in the supplementary materials:

1. Supplementary Figs. 7-8, Supplementary materials 2.2: The characterization and related discussions about mechanical reciprocity/nonreciprocity of the mechanical systems are added.

Reviewer #4 (Remarks to the Author):

Comments: The authors present a droplet-based transducer with asymmetric mechanical response. I carefully read the paper and the correspondence with the previous reviewers, and I tend to agree with the opinion of Reviewer 3 that this paper, while reporting interesting experiments, shows a poor understanding of the principle of mechanical reciprocity, and appears to oversell the reported results. It is surprising to me that the authors replied to the criticism of Reviewer 3 citing the Maxwell-Betti theorem, and referencing to the recent paper in Science, 2023, 380, 192-198, even copying a figure from that paper, which indeed showcases a nonreciprocal response. On the contrary, from what I can tell none of the results reported by the authors appears to demonstrate a nonreciprocal response. If the authors want to connect their results to a reciprocity-breaking device, they should indeed perform a measurement similar to the one they report from Fig. R11 in their own report, demonstrating that opposite forces applied to opposite positions of the sample lead to a different displacement. This would be a measurement of the degree of nonreciprocity of their mechanical system. It would also be good to compare the degree of broken nonreciprocity to the amplitude of the applied force, and to the recently reported systems that produce these effects, to demonstrate the effectiveness of their droplet-based setup. As it is now, the paper does not demonstrate nonreciprocal responses, hence the many references to reciprocity in the text are unwarranted. I do not recommend publication given this important deficiency.

Reply: We deeply thank the reviewer for the time and effort in thoroughly reading our manuscript and the previous correspondences. Following the reviewer's comments, we have carefully reviewed the literature to clarify the meaning of mechanical non-reciprocity, and find that the term is indeed misused in our previous manuscript. The term "mechanical non-reciprocity" refers to the difference in the ease of material deformation in two opposite directions (Science, 380, 192-198), rather than the asymmetry of droplet morphology change reported in our manuscript. Therefore, we have revised the manuscript and used the term "transformation symmetry/asymmetry" instead of "mechanical reciprocity/nonreciprocity" to describe the ability to transform a symmetric input into symmetric/asymmetric outputs. We are very appreciative of the reviewers for pointing out this problem.

According to the reviewer's comment, we have measured the nonreciprocity of the mechanical system by comparing the ease of deformation of the single-droplet system in opposite directions. We rotate the substrate in clockwise and anticlockwise directions to observe the droplet deformations under the same external gravity. As shown in Figure R3, when tilting the substrate with a circle pattern 90 degrees anticlockwise, the droplet displays a displacement of 0.41 mm, which is the same as the clockwise substrate. On the contrary, the deformations of the droplet are significantly different when tilting the substrate with a gourd pattern. Compared to the 3.82 mm displacement for the anticlockwise substrate, the droplet on the clockwise substrate displays a much smaller displacement of only 0.78 mm. This result suggests the existence of nonreciprocity in the droplet system.

Figure R3 Reciprocity characterization of the single-droplet system. (a) Droplet on a circle pattern shows mechanical reciprocity. (b) Droplet on a gourd pattern shows mechanical nonreciprocity.

In addition, we have quantitatively studied the relationship between the deformation extent of the droplet and the exerted force. The deforming force of the droplet is balanced by the surface tension ($\sim\gamma R$, where γ is the surface tension and R is the radius of the droplet), yielding a scale of 10^{-4} N, which is difficult to measure with a commercial force sensor. Alternatively, a thin polyethylene terephthalate sheet with a thickness of $70\ \mu\text{m}$ is utilized to act as a cantilever beam to deform the droplet and measure the force, as shown in Figure R4. The cantilever beam laps on the edge of the upper plate, and will bend to generate leftward force and deform the droplets if the beam moves to the left. The leftward force can be obtained by the deflection equation of the cantilever beam, which is:

$$F = \frac{Ebh^3W}{4l^3}$$

where elastic modulus $E = 2000$ MPa, beam width $b = 1$ mm, beam thickness $h = 0.07$ mm, beam length $l = 10.35$ mm, and W is the deflection of the cantilever beam.

Figure R4 Experimental facility for measuring the deforming force with a cantilever beam.

As shown in Figure R5a-b, the mechanical transducer with circle-arrayed pattern shows mechanical reciprocity because the same shear force induces the same displacement when pushing the upper plate to move to the left or right. However, for the mechanical transducer with gourd-arrayed pattern, a force of 0.4 mN can move the upper plate 1.05 mm to the left, but only a displacement of 0.65 mm is achieved to the right (Figure R5c-d). The differentiated ease of deformation in opposite directions arises from the asymmetry of the gourd pattern and the droplet morphology, concluding the existence of mechanical nonreciprocity in the mechanical transducer with gourd-arrayed pattern. This degree of nonreciprocity is lower than the results reported in the literature (Science, 2023, 380, 192-198), but the force used in the literature is 0.8 N, which is much larger than that in our study (0.4 mN), and the nonreciprocal behavior is difficult to be observed when the same magnitude of force as ours was used. This implies that our mechanical transducer may have a higher sensitivity. Considering that the findings of our study is more related to changes in the morphology of the droplets, especially the height variation, we agree that the asymmetry is more appropriate than non-reciprocity in our study, even though the non-reciprocal behavior is also found in the droplet-based mechanical transducer. The results and discussion about droplet mechanical reciprocity are added in the Supplementary Materials. We thanks very much to the reviewer for the useful guidance.

Figure R5 Mechanical reciprocity test of the droplet-arrayed mechanical system. (a-b) For the mechanical transducer with a circle array, the same shear force induces the same displacement to the left or right, showing mechanical reciprocity. (c-d) For the mechanical transducer with a gourd array, the same shear force induces different displacements to the left or right, showing mechanical nonreciprocity.

Modifications in the manuscript:

1. The title of the manuscript is revised as “Droplet-based mechanical transducers modulated by the symmetry of wettability patterns”.
2. Page 1, Lines 31-37: the sentences “From pistons driving gears ...” are added to illustrate the importance of mechanical energy conversion.
3. Page 2, Lines 43-44; Page 3, Lines 16-17; Page 4, Lines 19-20, 24, 35: the term “symmetric/asymmetric output” is used to describe the property of output vibration.
4. Page 1, Lines 12, 21; Page 4, Lines 8, 10, 39-40; Page 5, Line 15: the term “transformation symmetry/asymmetry” is used to describe the ability of the mechanical transducer to transform a symmetric input into symmetric/asymmetric outputs.
5. Page 4, Lines 35-38: The measurement result of the droplet reciprocity/nonreciprocity are added.

Modifications in the supplementary materials:

1. Supplementary Figs. 7-8, Supplementary materials 2.2: The characterization and related discussions about mechanical reciprocity/nonreciprocity of the mechanical systems are added.

REVIEWERS' COMMENTS

Reviewer #4 (Remarks to the Author):

The authors have for the most part addressed my concerns regarding the misuse of the term 'nonreciprocity'. My feeling reading their response is that they still do not properly use this term, and I recommend them to remove the sentence 'Interestingly, the droplet on the gourd pattern also shows mechanical nonreciprocity, as the asymmetry of pattern and droplet morphology cause differentiated ease of deformation in opposite directions.'

Once again, I stress that mechanical reciprocity (Maxwell-Betti theorem) implies that, if a force applied at point A generates a displacement at point B, reciprocity requires that the same force applied at point B would generate the same displacement at point A. I do not see this experiment performed in this paper, and I understand from the authors' response that this cannot be done in the present setup. If this is the case, the authors should not refer to mechanical nonreciprocity, and remove the sentence above from the paper. The authors show that the same force, when applied in opposite directions, produces a different deformation. This is not a nonreciprocal response, and can be achieved without violating Maxwell-Betti theorem in an asymmetric device.

Reviewer #4 (Remarks to the Author):

The authors have for the most part addressed my concerns regarding the misuse of the term 'nonreciprocity'. My feeling reading their response is that they still do not properly use this term, and I recommend them to remove the sentence 'Interestingly, the droplet on the gourd pattern also shows mechanical nonreciprocity, as the asymmetry of pattern and droplet morphology cause differentiated ease of deformation in opposite directions.'

Once again, I stress that mechanical reciprocity (Maxwell-Betti theorem) implies that, if a force applied at point A generates a displacement at point B, reciprocity requires that the same force applied at point B would generate the same displacement at point A. I do not see this experiment performed in this paper, and I understand from the authors' response that this cannot be done in the present setup. If this is the case, the authors should not refer to mechanical nonreciprocity, and remove the sentence above from the paper. The authors show that the same force, when applied in opposite directions, produces a different deformation. This is not a nonreciprocal response, and can be achieved without violating Maxwell-Betti theorem in an asymmetric device.

Reply: We deeply thank the reviewer for the time and efforts in reading our manuscript again and providing constructive suggestions. According to the reviewer's suggestion, we have removed the sentence "Interestingly, the droplet on the gourd pattern also shows mechanical nonreciprocity, as the asymmetry of pattern and droplet morphology cause differentiated ease of deformation in opposite directions". As it is difficult to apply a force to a specific point of the droplet and measure the displacement of another point due to the flowability of droplets, we decided not to refer to mechanical nonreciprocity in our manuscript. We are very grateful to the reviewer for pointing out the problem, which has greatly benefited us.

Modifications in the manuscript:

The statement about mechanical reciprocity ("Interestingly, the droplet on the gourd pattern also shows mechanical nonreciprocity, ...") has been removed.

Modifications in the supplementary materials:

The discussion about mechanical reciprocity (Supplementary Figures 7 and 8, Supplementary Text 2.2 in the previous version) has been removed.